# LEXTREME: A Multi-Lingual and Multi-Task Benchmark for the Legal Domain

**Joel Niklaus** [1,2,6*]   **Veton Matoshi** [2*]   **Pooja Rani** [3]

**Andrea Galassi** [4]   **Matthias Stürmer** [1,2]   **Ilias Chalkidis** [5]

[1]University of Bern   [2]Bern University of Applied Sciences   [3]University of Zurich
[4]University of Bologna   [5]University of Copenhagen   [6]Stanford University

## Abstract

Lately, propelled by phenomenal advances around the transformer architecture, the legal NLP field has enjoyed spectacular growth. To measure progress, well-curated and challenging benchmarks are crucial. Previous efforts have produced numerous benchmarks for general NLP models, typically based on news or Wikipedia. However, these may not fit specific domains such as law, with its unique lexicons and intricate sentence structures. Even though there is a rising need to build NLP systems for languages other than English, many benchmarks are available only in English and no multilingual benchmark exists in the legal NLP field. We survey the legal NLP literature and select 11 datasets covering 24 languages, creating LEXTREME. To fairly compare models, we propose two aggregate scores, i.e., dataset aggregate score and language aggregate score. Our results show that even the best baseline only achieves modest results, and also ChatGPT struggles with many tasks. This indicates that LEXTREME remains a challenging task with ample room for improvement. To facilitate easy use for researchers and practitioners, we release LEXTREME on huggingface along with a public leaderboard and the necessary code to evaluate models. We also provide a public Weights and Biases project containing all runs for transparency.

## 1 Introduction

In the last decade, Natural Language Processing (NLP) has gained relevance in Legal Artificial Intelligence, transitioning from symbolic to subsymbolic techniques (Villata et al., 2022). Such a shift is motivated partially by the nature of legal resources, which appear primarily in a textual format (legislation, legal proceedings, contracts, etc.). Following the advancements in NLP technologies, the legal NLP literature (Zhong et al., 2020; Aletras et al., 2022; Katz et al., 2023) is flourishing

---

* Equal contribution.

**LEXTREME Score (mF1)**

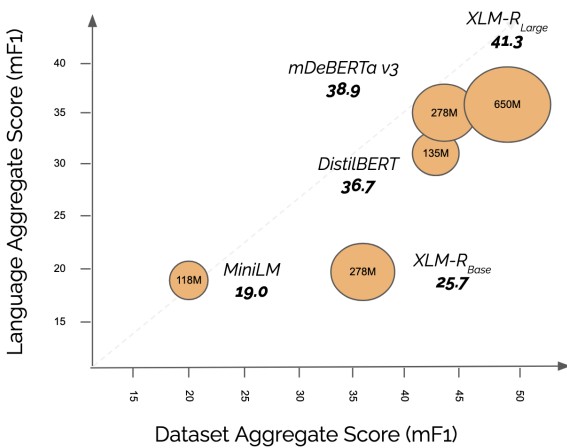

Figure 1: Overview of multilingual models on LEXTREME. The bubble size and text inside indicate the parameter count. The bold number below the model name indicates the LEXTREME score (harmonic mean of the language agg. score and the dataset agg. score).

with many new resources, such as large legal corpora (Henderson et al., 2022), task-specific datasets (Shen et al., 2022; Christen et al., 2023; Brugger et al., 2023; Niklaus et al., 2023a), and pre-trained legal-oriented language models (Chalkidis et al., 2020; Zheng et al., 2021; Xiao et al., 2021; Niklaus and Giofre, 2023; Hua et al., 2022; Chalkidis et al., 2023). Greco and Tagarelli (2023) offer a comprehensive survey on the topic.

Specifically, the emergence of pre-trained Language Models (PLMs) has led to significant performance boosts on popular benchmarks like GLUE (Wang et al., 2019b) or SuperGLUE (Wang et al., 2019a), emphasizing the need for more challenging benchmarks to measure progress. Legal benchmark suites have also been developed to systematically evaluate the performance of PLMs, showcasing the superiority of legal-oriented models over generic ones on downstream tasks such as legal document classification or question answering (Chalkidis et al., 2022a; Hwang et al., 2022).

Even though these PLMs are shown to be effective for numerous downstream tasks, they are general-purpose models that are trained on broad-domain resources, such as Wikipedia or News, and therefore, can be insufficient to address tasks specific to the legal domain (Chalkidis et al., 2020; Hua et al., 2022; Niklaus and Giofre, 2023). Indeed, the legal domain is strongly characterized both by its lexicon and by specific knowledge typically not available outside of specialized domain resources. Laypeople even sometimes call the language used in legal documents "legalese" or "legal jargon", emphasizing its complexity. Moreover, the length of a legal document usually exceeds the length of a Wikipedia or news article, and in some tasks the relationships between its entities may span across the entire document. Therefore, it is necessary to develop specialized Legal PLMs trained on extensive collections of legal documents and evaluate them on standardized legal benchmarks. While new PLMs capable of handling long documents have been developed in the last years, they are predominantly trained for the general domain and on English data only.

The rising need to build NLP systems for languages other than English, the lack of textual resources for such languages, and the widespread use of code-switching in many cultures (Torres Cacoullos, 2020) is pushing researchers to train models on massively multilingual data (Conneau et al., 2020). Nonetheless, to the best of our knowledge, no multilingual legal language model has been proposed so far. Consequently, there is a need for standardized multilingual benchmarks that can be used to evaluate existing models and assess whether more research efforts should be directed toward the development of domain-specific models. This is particularly important for legal NLP where inherently multinational (European Union, Council of Europe) or multilingual (Canada, Switzerland) legal systems are prevalent.

In this work, we propose a challenging multilingual benchmark for the legal domain, named LEXTREME. We survey the literature from 2010 to 2022 and select 11 relevant NLU datasets covering 24 languages in 8 subdivisions (Germanic, Romance, Slavic, Baltic, Greek, Celtic, Finnic, and Hungarian) from two language families (Indo-European and Uralic). We evaluate five widely used multilingual encoder-based language models as shown in Figure 1 and observe a correlation

between the model size and performance on LEXTREME. Surprisingly, at the low end, DistilBERT (Sanh et al., 2019) strongly outperforms MiniLM (Wang et al., 2020) (36.7 vs 19.0 LEXTREME score) while only having marginally more parameters (135M vs 118M).

For easy evaluation of future models, we release the aggregate dataset on the huggingface hub [1] along with a public leaderboard and the necessary code to run experiments on GitHub.[2] Knowing that our work can not encompass "Everything in the Whole Wide Legal World" (Raji et al., 2021), we design LEXTREME as a living benchmark and provide detailed guidelines on our repository and encourage the community to contribute high-quality multilingual legal datasets.[3] Finally, we integrated LEXTREME together with the popular English legal benchmark LexGLUE (Chalkidis et al., 2022a) into HELM (Liang et al., 2022) (an effort to evaluate language models holistically using a large number of datasets from diverse tasks) to ease the adoption of curated legal benchmarks also for the evaluation of large language models such as GPT-3 (Brown et al., 2020), PALM (Chowdhery et al., 2022) or LLaMA (Touvron et al., 2023).

**Contributions** of this paper are two-fold:

1. We review the legal NLP literature to find relevant legal datasets and compile a multilingual legal benchmark of 11 datasets in 24 languages from 8 language groups.
2. We evaluate several baselines on LEXTREME to provide a reference point for researchers and practitioners to compare to.

## 2 Related Work

### 2.1 Benchmarks

Benchmarking is an established method to enable easy and systematic comparison of approaches. GLUE (Wang et al., 2019b) is one of the first benchmarks to evaluate general-purpose neural language models. It is a set of supervised sentence understanding predictive tasks in English that were created through aggregation and curation of several existing datasets. However, it became quickly obsolete due to advanced contextual language models, such as BERT (Devlin et al., 2019), which

---

[1]https://huggingface.co/datasets/joelniklaus/lextreme
[2]https://github.com/JoelNiklaus/LEXTREME
[3]Since the release of this call in February 2023, already eleven new tasks have been contributed and integrated.

| Name | Source | Domain | Tasks | Datasets | Languages | Agg. Score |
|------|--------|--------|-------|----------|-----------|------------|
| GLUE | (Wang et al., 2019b) | Misc. Texts | 7 | 9 | English | Yes |
| SUPERGLUE | (Wang et al., 2019a) | Misc. Texts | 8 | 8 | English | Yes |
| MMLU | (Hendrycks et al., 2021) | Misc. Texts | 1 | 57 | English | Yes |
| CLUE | (Xu et al., 2020) | Misc. Texts | 9 | 9 | Chinese | Yes |
| XTREME | (Hu et al., 2020) | Misc. Texts | 6 | 9 | 40 | Yes |
| BLUE | (Peng et al., 2019) | Biomedical Texts | 5 | 10 | English | Yes |
| CBLUE | (Zhang et al., 2022) | Biomedical Texts | 9 | 9 | Chinese | Yes |
| LegalBench | (Guha et al., 2022) | Legal Texts | 44 | 8. | English | No |
| LexGLUE | (Chalkidis et al., 2022a) | Legal Texts | 7 | 6 | English | Yes |
| FairLex | (Chalkidis et al., 2022b) | Legal Texts | 4 | 4 | 5 | No |
| LBOX | (Hwang et al., 2022) | Legal Texts | 5 | 5 | Korean | Yes |
| LEXTREME | (our work) | Legal Texts | 18 | 11 | 24 | Yes |
| SUPERB | (Yang et al., 2021) | Speech | 10 | 10 | English | No |
| SUPERB-SG | (Tsai et al., 2022) | Speech | 5 | 5 | English | No |
| TAPE | (Rao et al., 2019) | Proteins | 5 | 5 | n/a | No |

Table 1: Characteristics of popular existing NLP benchmarks.

excelled on most tasks. Subsequently, its updated version, named SUPERGLUE (Wang et al., 2019a) was proposed, incorporating new predictive tasks that are solvable by humans but are difficult for machines. Both benchmarks proposed an evaluation score computed as an aggregation of the scores obtained by the same model on each task. They are also agnostic regarding the pre-training of the model, and do not provide a specific corpus for it. Inspired by these works, numerous benchmarks have been proposed over the years. We describe some well-known ones in Table 1.

The MMLU benchmark is specifically designed to evaluate the knowledge acquired during the pre-training phase of the model by featuring only zero-shot and few-shot learning tasks (Hendrycks et al., 2021). Similarly, SUPERB (Yang et al., 2021) and SUPERB-SG (Tsai et al., 2022) were proposed for speech data, unifying well-known datasets. However, they mainly vary in tasks, e.g., SUPERB-SG includes both predictive and generative tasks, which makes it different from the other benchmarks discussed in this section. Additionally, SUPERB-SG includes diverse tasks, such as speech translation and cross-lingual automatic speech recognition, which require knowledge of languages other than English. Neither of the two (SUPERB or SUPERB-SG) proposes an aggregated score.

XTREME (Hu et al., 2020) is a benchmark specifically designed to evaluate the ability of cross-lingual generalization of models. It includes six cross-lingual predictive tasks over ten datasets of miscellaneous texts, covering a total of 40 languages. While some original datasets in it were al-

ready designed for cross-lingual tasks, others were extended by translating part of the data using human professionals and automatic methods.

## 2.2 Benchmarks for the Legal Domain

LEXGLUE (Chalkidis et al., 2022a) is the first benchmark for the legal domain and covers six predictive tasks over five datasets made of textual documents in English from the US, EU, and Council of Europe. While some tasks may not require specific legal knowledge to be solved, others would probably need, or at least benefit from, information regarding the EU or US legislation on the specific topic. Among the main limitations of their benchmark, Chalkidis et al. highlight its monolingual nature and remark that "*there is an increasing need for developing models for other languages*". In parallel, Chalkidis et al. (2022b) released FairLex, a multilingual benchmark for the evaluation of fairness in legal NLP tasks. With a similar aim, Hwang et al. (2022) released the LBOX benchmark, covering two classification tasks, two legal judgment prediction tasks, and one Korean summarization task. Motivated by LEXGLUE and LBOX, we propose a benchmark to encourage multilingual models, diverse tasks, and datasets for the legal domain. Guha et al. (2022) proposed the LEGAL-BENCH initiative that aims to establish an open and collaborative legal reasoning benchmark for few-shot evaluation of LLMs where legal practitioners and other domain experts can contribute by submitting tasks. At its creation, the authors have already added 44 lightweight tasks. While most tasks require legal reasoning based on the common

law system (mostly prevalent in the UK and former colonies), there is also a clause classification task. For a more comprehensive overview of the many tasks related to automated legal text analysis, we recommend reading the works of Chalkidis et al. (2022a) and Zhong et al. (2020).

## 2.3 Legal Language Models

Several works have proposed legal language models (models specifically trained for the legal domain) for several languages other than English. For example, legal language models for English (Chalkidis et al., 2020; Yin and Habernal, 2022; Chalkidis et al., 2023), French (Douka et al., 2021), Romanian (Masala et al., 2021), Italian (Tagarelli and Simeri, 2022; Licari and Comandé, 2022), Chinese (Xiao et al., 2021), Arabic (Al-Qurishi et al., 2022), Korean (Hwang et al., 2022), and Portuguese (Ciurlino, 2021). Recently, pre-trained multilingual legal language models (Niklaus et al., 2023b; Rasiah et al., 2023) have been released. Unfortunately, these models were not available at the time of submission, so we do not present results as part of this work.

## 3 LEXTREME Datasets and Tasks

### 3.1 Dataset and Task Selection

To find relevant datasets for the LEXTREME benchmark we explore the literature of NLP and the legal domain, identifying relevant venues such as ACL, EACL, NAACL, EMNLP, LREC, ICAIL, and the NLLP workshop. We search the literature on these venues for the years 2010 to 2022. We search for some common keywords (case insensitive) that are related to the legal domain, e.g., *criminal*, *judicial*, *judgment*, *jurisdictions*, *law*, *legal*, *legislation*, and dataset, e.g., *dataset*, and *corpus* vie their union. These keywords help to select 108 potentially relevant papers. Then, we formulate several criteria to select the datasets. Finally, three authors analyze the candidate papers and perform the selection. We handled the disagreement between authors based on mutual discussion and the majority voting mechanism.

**Inclusion criteria:**

I1: It is about legal text (e.g., patents are not considered part of the legal text)

I2: It performs legal tasks (e.g., judgment prediction) or NLU tasks on legal text in order to have datasets that understand or reason

| Task | # Examples | # Labels |
|------|-----------|----------|
| BCD-J | 3234 / 404 / 405 | 3 / 3 / 3 |
| BCD-U | 1715 / 211 / 204 | 2 / 2 / 2 |
| GAM | 19271 / 2726 / 3078 | 4 / 4 / 4 |
| GLC-V | 28536 / 9511 / 9516 | 47 / 47 / 47 |
| GLC-C | 28536 / 9511 / 9516 | 386 / 377 / 374 |
| GLC-S | 28536 / 9511 / 9516 | 2143 / 1679 / 1685 |
| SJP | 59709 / 8208 / 17357 | 2 / 2 / 2 |
| OTS-UL | 2074 / 191 / 417 | 3 / 3 / 3 |
| OTS-CT | 19942 / 1690 / 4297 | 9 / 8 / 9 |
| C19 | 3312 / 418 / 418 | 8 / 8 / 8 |
| MEU-1 | 817239 / 112500 / 115000 | 21 / 21 / 21 |
| MEU-2 | 817239 / 112500 / 115000 | 127 / 126 / 127 |
| MEU-3 | 817239 / 112500 / 115000 | 500 / 454 / 465 |
| GLN | 17699 / 4909 / 4017 | 17 / 17 / 17 |
| LNR | 7552 / 966 / 907 | 11 / 9 / 11 |
| LNB | 7828 / 1177 / 1390 | 13 / 13 / 13 |
| MAP-C | 27823 / 3354 / 10590 | 13 / 11 / 11 |
| MAP-F | 27823 / 3354 / 10590 | 44 / 26 / 34 |

Table 2: Dataset and task overview. *# Examples* and *# Labels* show values for train, validation, and test splits.

about the legal text, similar to LEXGLUE (Chalkidis et al., 2022a)

I3: The current tasks are set in a European language, as per the scope of our present work, but we aim to incorporate a broader range of languages in future iterations of LEXTREME

I4: The dataset is annotated by humans directly or indirectly (e.g., judgement labels are extracted with regexes)

**Exclusion criteria:**

E1: The dataset is not publicly available

E2: The dataset does not contain a public license or does not allow data redistribution

E3: The dataset contains labels generated with ML systems

E4: It is not a peer-reviewed paper

After applying the above criteria, we select 11 datasets from 108 papers. We provide the list of all these datasets in our repository.

### 3.2 LEXTREME Datasets

In the following, we briefly describe the selected datasets. Table 2 provides more information about the number of examples and label classes per split for each task. For a detailed overview of the jurisdictions as well as the number of languages covered by each dataset, see Table 3. Each dataset can have several configurations (tasks), which are the basis of our analyses, i.e., the pre-trained models have always been fine-tuned on a single task. LEXTREME consists of three task types: Single

| Dataset | Jurisdiction | Languages |
|---------|--------------|-----------|
| BCD | BR | pt |
| GAM | DE | de |
| GLC | GR | el |
| SJP | CH | de, fr, it |
| OTS | EU | de, en, it, pl |
| C19 | BE, FR, HU, IT, NL, PL, UK | en, fr, hu, it, nb, nl, pl |
| MEU | EU | 24 EU langs |
| GLN | GR | el |
| LNR | RO | ro |
| LNB | BR | pt |
| MAP | EU | 24 EU langs |

Table 3: Overview of datasets, the jurisdiction, and the languages. The 24 EU languages are: bg, cs, da, de, el, en, es, et, fi, fr, ga, hu, it, lt, lv, mt, nl, pt, ro, sk, sv.

Label Text Classification (SLTC), Multi Label Text Classification (MLTC), and Named Entity Recognition (NER). We use the existing train, validation, and test splits if present. Otherwise, we split the data randomly ourselves (80% train, 10% validation, and 10% test).

**Brazilian Court Decisions (BCD).** Legal systems are often huge and complex, and the information is scattered across various sources. Thus, predicting case outcomes from multiple vast volumes of litigation is a difficult task. Lage-Freitas et al. (2022) propose an approach to predict Brazilian legal decisions to support legal practitioners. We use their dataset from the State Supreme Court of Alagoas (Brazil). The input to the models is always the case description. We perform two SLTC tasks: In the BCD-J subset models predict the approval or dismissal of the case or appeal with the three labels *no, partial, yes*, and in the BCD-U models predict the judges' unanimity on the decision alongside two labels, namely *unanimity, not-unanimity*.

**German Argument Mining (GAM).** Identifying arguments in court decisions is vital and challenging for legal practitioners. Urchs. et al. (2021) assembled a dataset of 200 German court decisions for sentence classification based on argumentative function. We utilize this dataset for a SLTC task. Model input is a sentence, and output is categorized as *conclusion*, *definition*, *subsumption*, or *other*.

**Greek Legal Code (GLC).** Legal documents can cover a wide variety of topics, which makes accurate topic classification all the more important. Papaloukas et al. (2021) compiled a dataset for topic classification of Greek legislation documents. The

documents cover 47 main thematic topics which are called *volumes*. Each of them is divided into thematic sub categories which are called *chapters* and subsequently, each chapter breaks down to *subjects*. Therefore, the dataset is used to perform three different SLTC tasks along volume level (GLC-V), chapter level (GLC-C), and subject level (GLC-S). The input to the models is the entire document, and the output is one of the several topic categories.

**Swiss Judgment Prediction (SJP).** Niklaus et al. (2021, 2022), focus on predicting the judgment outcome of 85K cases from the Swiss Federal Supreme Court (FSCS). The input to the models is the appeal description, and the output is whether the appeal is approved or dismissed (SLTC task).

**Online Terms of Service (OTS).** While multilingualism's benefits (e.g., cultural diversity) in the EU legal world are well-known (Commission, 2005), creating an official version of every legal act in 24 languages raises interpretative challenges. Drawzeski et al. (2021) attempt to automatically detect unfair clauses in terms of service documents. We use their dataset of 100 contracts to perform a SLTC and MLTC task. For the SLTC task (OTS-UL), model inputs are sentences, and outputs are classifications into three unfairness levels: *clearly fair*, *potentially unfair* and *clearly unfair*. The MLTC task (OTS-CT) involves identifying sentences based on nine clause topics.

**COVID19 Emergency Event (C19).** The COVID-19 pandemic showed various exceptional measures governments worldwide have taken to contain the virus. Tziafas et al. (2021), presented a dataset, also known as EXCEPTIUS, that contains legal documents with sentence-level annotation from several European countries to automatically identify the measures. We use their dataset to perform the MLTC task of identifying the type of measure described in a sentence. The input to the models are the sentences, and the output is neither or at least one of the measurement types.

**MultiEURLEX (MEU).** Multilingual transfer learning has gained significant attention recently due to its increasing applications in NLP tasks. Chalkidis et al. (2021a) explored the cross-lingual transfer for legal NLP and presented a corpus of 65K EU laws annotated with multiple labels from the EUROVOC taxonomy. We perform a MLTC task to identify labels (given in the taxonomy) for

each document. Since the taxonomy exists on multiple levels, we prepare configurations according to three levels (MEU-1, MEU-2, MEU-3).

**Greek Legal NER (GLN).** Identifying various named entities from natural language text plays an important role for Natural Language Understanding (NLU). Angelidis et al. (2018) compiled an annotated dataset for NER in Greek legal documents. The source material are 254 daily issues of the Greek Government Gazette over the period 2000-2017. In *all* NER tasks of LEXTREME the input to the models is the list of tokens, and the output is an entity label for each token.

**LegalNERo (LNR).** Similar to GLN, Pais et al. (2021) manually annotated Romanian legal documents for various named entities. The dataset is derived from 370 documents from the larger MAR-CELL Romanian legislative subcorpus.[4]

**LeNER BR (LNB).** Luz de Araujo et al. (2018) compiled a dataset for NER for Brazilian legal documents. 66 legal documents from several Brazilian Courts and four legislation documents were collected, resulting in a total of 70 documents annotated for named entities.

**MAPA (MAP).** de Gibert et al. (2022) built a multilingual corpus based on EUR-Lex (Baisa et al., 2016) for NER annotated at a coarse-grained (MAP-C) and fine-grained (MAP-F) level.

## 4 Models Considered

Since our benchmark only contains NLU tasks, we consider encoder-only models for simplicity. Due to resource constraints, we did not fine-tune models larger than 1B parameters.

### 4.1 Multilingual

We considered the five multilingual models listed in Table 4, trained on at least 100 languages each (more details are in Appendix B). For XLM-R we considered both the base and large version. Furthermore, we used ChatGPT (gpt-3.5-turbo) for zero-shot evaluation of the text classification tasks with less than 50 labels.[5] To be fair across tasks we did not consider few-shot evaluation or more

sophisticated prompting techniques because of prohibitively long inputs in many tasks.

### 4.2 Monolingual

In addition to the multilingual models, we also fine-tuned available monolingual models on the language specific subsets. We chose monolingual models only if a certain language was represented in at least three datasets.[6] We made a distinction between general purpose models, i.e., models that have been pre-trained on generic data aka *Native-BERTs*, and legal models, i.e., models that have been trained (primarily) on legal data aka *NativeLegalBERTs*. A list of the monolingual models can be found in the appendix in Table 8.

### 4.3 Hierarchical Variants

A significant part of the datasets consists of very long documents, the best examples being all variants of MultiEURLEX; we provide detailed using different tokenizers on all datasets in our online repository. However, the models we evaluated were all pre-trained with a maximum sequence length of only 512 tokens. Directly applying pre-trained language models on lengthy legal documents may necessitate substantial truncation, severely restricting the models. To overcome this limitation, we use hierarchical versions of pre-training models for datasets containing long documents.

The hierarchical variants used here are broadly equivalent to those in (Chalkidis et al., 2021b; Niklaus et al., 2022). First, we divide each document into chunks of 128 tokens each. Second, we use the model to be evaluated to encode each of these paragraphs in parallel and to obtain the [CLS] embedding of each chunk which can be used as a context-unaware chunk representation. In order to make them context-aware, i.e. aware of the surrounding chunks, the chunk representations are fed into a 2-layered transformer encoder. Finally, max-pooling over the context-aware paragraph representations is deployed, which results in a document representation that is fed to a classification layer.

Unfortunately, to the best of our knowledge models capable of handling longer context out of the box, such as Longformer (Beltagy et al., 2020) and SLED (Ivgi et al., 2023) are not available multilingually and predominantly trained on English data only.

---

[4]https://marcell-project.eu/deliverables.html

[5]We excluded MultiEurlex because it only contains numeric labels and not textual ones, and because the inputs are very long in 24 languages rendering a valid comparison with reasonable costs impossible.

---

[6]Which is why we did not include Norwegian pre-trained models, even though Norwegian is covered in C19.

| Model | Source | # Parameters | Vocab | # Steps | Batch Size | Corpus | # Langs |
|-------|--------|-------------:|-------|--------:|-----------:|--------|--------:|
| MiniLM | Wang et al. (2020) | 118M | 250K | 1M | 256 | 2.5TB CC100 | 100 |
| DistilBERT | Sanh et al. (2019) | 135M | 120K | n/a | < 4000 | Wikipedia | 104 |
| mDeBERTa-v3 | He et al. (2020, 2021) | 278M | 128K | 500K | 8192 | 2.5TB CC100 | 100 |
| XLM-R$_{Base/Large}$ | Conneau et al. (2020) | 278M/560M | 250K | 1.5M | 8192 | 2.5TB CC100 | 100 |

Table 4: Multilingual models. All models support a maximum sequence length of 512 tokens. The third column shows the total number of parameters, including the embedding layer.

## 5 Experimental Setup

Multilingual models were fine-tuned on all languages of specific datasets. Monolingual models used only the given model's language subset.

Some datasets are highly imbalanced, one of the best examples being BCD-U with a proportion of the minority class of about 2%. Therefore, we applied random oversampling on all SLTC datasets, except for GLC, since all its subsets have too many labels, which would have led to a drastic increase in the data size and thus in the computational costs for fine-tuning. For each run, we used the same hyperparameters, as described in Section A.3.

As described in Section 4.3, some tasks contain very long documents, requiring the usage of hierarchical variants to process sequence lenghts of 1024 to 4096 tokens. Based on the distribution of the sequence length per example for each task (cf. Appendix H), we decided on suitable sequence lengths for each task before fine-tuning. A list of suitable sequence lengths are in A.1.

### 5.1 Evaluation Metrics.

We use the macro-F1 score for all datasets to ensure comparability across the entire benchmark, since it can be computed for both text classification and NER tasks. Mathew's Correlation Coefficient (MCC) (Matthews, 1975) is a suitable score for evaluating text classification tasks but its applicability to NER tasks is unclear. For brevity, we do not display additional scores, but more detailed (such as precision and recall, and scores per seed) and additional scores (such as MCC) can be found online on our Weights and Biases project.[7]

### 5.2 Aggregate Score

We acknowledge that the datasets included in LEXTREME are diverse and hard to compare due to variations in the number of samples and task complexity (Raji et al., 2021). This is why we always report the scores for each dataset subset, enabling a

---

[7] https://wandb.ai/lextreme/paper_results

fine-grained analysis. However, we believe that by taking the following three measures, an aggregate score can provide more benefits than drawbacks, encouraging the community to evaluate multilingual legal models on a curated benchmark, thus easing comparisons.

We (a) evaluate all datasets with the same score (macro-F1) making aggregation more intuitive and easier to interpret, (b) aggregating the F1 scores again using the harmonic mean, since F1 scores are already rates and obtained using the harmonic mean over precision and recall, following Tatiana and Valentin (2021), and (c) basing our final aggregate score on two intermediate aggregate scores — the dataset aggregate and language aggregate score – thus weighing datasets and languages equally promoting model fairness, following Tatiana and Valentin (2021) and Chalkidis et al. (2022a).

The final LEXTREME score is computed using the harmonic mean of the dataset and the language aggregate score. We calculate the dataset aggregate by successively taking the harmonic mean of (i) the languages in the configurations (e.g., de,fr,it in SJP), (ii) configurations within datasets (e.g., OTS-UL, OTS-CT in OTS), and (iii) datasets in LEXTREME (BCD, GAM). The language aggregate score is computed similarly: by taking the harmonic mean of (i) configurations within datasets, (ii) datasets for each language (e.g., MAP, MEU for lv), and (iii) languages in LEXTREME (bg,cs).

We do not address the dimension of the jurisdiction, which we consider beyond the scope of this work.

## 6 Results

In this section, we discuss baseline evaluations. Scores and standard deviations for validation and test datasets across seeds are on our Weights and Biases project or can be found in Table 11, 12, 13, 14. Comparisons with prior results on each dataset can be drawn from the tables provdided in section G in the appendix. Aggregated results by dataset and language are in Tables 5 and 6.

| Model | BCD | GAM | GLC | SJP | OTS | C19 | MEU | GLN | LNR | LNB | MAP | Agg. |
|---|---|---|---|---|---|---|---|---|---|---|---|---|
| MiniLM | 52.0 | **73.3** | 12.3 | 67.7 | 21.8 | 4.5 | 12.2 | 43.5 | 46.4 | 86.1 | 52.9 | 19.9 |
| DistilBERT | 53.7 | 69.5 | 53.4 | 66.8 | 52.4 | 21.2 | 23.2 | 38.1 | 48.0 | 78.7 | 53.0 | 43.2 |
| mDeBERTa v3 | 59.1 | 71.3 | 26.5 | 69.1 | 63.7 | **26.4** | 24.7 | 44.8 | 46.7 | 87.3 | **58.6** | 44.1 |
| XLM-R$_{Base}$ | **62.6** | 71.9 | 42.1 | **69.3** | 64.6 | 18.4 | 11.4 | **46.4** | 45.6 | 87.3 | 53.2 | 36.8 |
| XLM-R$_{Large}$ | 58.0 | 73.1 | **71.7** | 68.9 | **73.8** | 20.6 | **27.9** | 45.1 | **55.4** | **88.4** | 55.1 | **48.5** |

Table 5: Dataset aggregate scores for multilingual models. The best scores are in bold.

| Model | bg | cs | da | de | el | en | es | et | fi | fr | ga | hr | hu | it | lt | lv | mt | nl | pl | pt | ro | sk | sl | sv | Agg. |
|---|---|---|---|---|---|---|---|---|---|---|---|---|---|---|---|---|---|---|---|---|---|---|---|---|---|
| MiniLM | 20.9 | 20.4 | 19.8 | 27.6 | 19.0 | 8.2 | 21.2 | 19.9 | 19.7 | 15.9 | 40.2 | 11.9 | 20.3 | 15.1 | 20.3 | 20.3 | 14.9 | 20.5 | 15.2 | 30.5 | 25.9 | 20.2 | 12.4 | 20.5 | 18.1 |
| DistilBERT | 33.7 | 32.7 | 32.1 | 47.4 | 35.1 | 38.0 | 36.1 | 31.0 | 30.8 | 38.8 | 43.8 | 22.5 | 31.5 | 41.0 | 31.7 | 31.6 | 29.9 | 19.0 | 25.0 | 44.5 | 37.6 | 32.0 | 22.9 | 33.3 | 31.9 |
| mDeBERTa v3 | 34.6 | 34.4 | 33.6 | 49.9 | 33.5 | 41.0 | 36.6 | 34.6 | 33.9 | 39.8 | **49.4** | 24.7 | 35.6 | 44.5 | 34.9 | 35.0 | 33.4 | 24.5 | 29.2 | 46.3 | 39.5 | 35.6 | 24.8 | 35.9 | 34.8 |
| XLM-R$_{Base}$ | 19.9 | 19.4 | 18.8 | 33.3 | 25.6 | 27.6 | 19.4 | 18.8 | 18.6 | 27.3 | 44.9 | 11.6 | 13.4 | 31.0 | 18.7 | 18.9 | 16.0 | 15.2 | 21.4 | 30.3 | 24.1 | 18.9 | 11.7 | 19.4 | 19.7 |
| XLM-R$_{Large}$ | **38.5** | 37.9 | 38.0 | 51.5 | 44.1 | 44.7 | 39.7 | 36.9 | 35.3 | 42.1 | 48.6 | **28.1** | 22.7 | **48.0** | 37.4 | 37.9 | 34.1 | 19.3 | 32.7 | 48.9 | 42.3 | **37.1** | **28.0** | 37.0 | 36.0 |
| NativeLegalBERT | - | - | - | - | - | 43.8 | **40.3** | - | - | - | - | - | 34.0 | - | - | - | - | - | - | 38.8 | - | - | - | - | 38.9 |
| NativeBERT | 24.3 | **47.4** | 42.8 | 56.0 | 47.9 | 49.4 | 33.3 | 38.3 | 43.2 | 43.5 | 44.0 | - | 45.4 | 42.5 | - | - | - | 36.2 | 21.6 | 54.9 | 44.4 | 29.1 | - | 46.1 | 39,1 |

Table 6: Language aggregate scores for multilingual models. The best scores are in bold. For each language, we also list the best-performing monolingual legal model under *NativeLegalBERT* and the best-performing monolingual non-legal model under *NativeBERT*. Missing values indicate that no suitable models were found.

**Larger models are better**   For both, we see a clear trend that larger models perform better. However, when looking at the individual datasets and languages, the scores are more erratic. Note that XLM-R$_{Base}$ underperforms on MEU (especially on MEU-3; see Table 11 and Table 12) leading to a low dataset aggregate score due to the harmonic mean. Additionally, low performance on MEU-3 has a large impact on its language aggregate score, since it affects all 24 languages.

**Differing model variance across datasets**   We observe significant variations across datasets such as GLC, OTS or C19, with differences as large as 52 (in OTS) between the worst-performing MiniLM and the best-performing XLM-R large. MiniLM seems to struggle greatly with these three datasets, while even achieving the best performance on GAM. On other datasets, such as GAM, SJP, and MAP the models are very close together (less than 6 points between best and worst model). Even though XLM-R$_{Large}$ takes the top spot on aggregate, it only has the best performance in six out of eleven datasets.

**Less variability across languages**   In contrast to inconsistent results on the datasets, XLM-R$_{Large}$ outperforms the other multilingual models on most languages (21 out of 24). Additionally, we note that model variability within a language is similar to the variability within a dataset, however, we don't see extreme cases such as GLC, OTS, or C19.

**Monolingual models are strong**   Monolingual general-purpose models (NativeBERT in Table 6)

| Task | XLM-R$_{Large}$ | ChatGPT |
|---|---|---|
| BCD-J | **58.1** | 52.1 |
| BCD-U | **70.4** | 48.2 |
| GAM | **73.0** | 35.5 |
| GLC-V | **58.2** | 32.9 |
| SJP | **60.9** | 51.2 |
| OTS-UL | **79.8** | 15.1 |
| OTS-CT | **64.5** | 12.7 |
| C19 | **27.7** | 23.6 |

Table 7: Results with ChatGPT on the validation sets performed on June 15, 2023. Best results are in **bold**.

show strong performance with only a few exceptions (on Bulgarian, Spanish, Polish, and Slovak). In 13 out of 19 available languages they reach the top performance, leading to the top language aggregate score. The few available models pre-trained on legal data (NativeLegalBERT) slightly outperform multilingual models of the same size.

**ChatGPT underperforms**   We show a comparison of ChatGPT with the best performing multilingual model XLM-R$_{Large}$ in Table 7. To save costs, we limited the evaluation size to 1000 samples for ChatGPT. We use the validation set instead of the test set to be careful not to leak test data into ChatGPT, possibly affecting future evaluation. Chalkidis (2023) showed that ChatGPT is still outperformed by supervised approaches on LexGLUE. Similarly, we find that much smaller supervised models clearly outperform ChatGPT in all of tested tasks, with very large gaps in GAM and OTS.

## 7 Conclusions and Future Work

**Conclusions** We survey the literature and select 11 datasets out of 108 papers with rigorous criteria to compile the first multilingual benchmark for legal NLP. By open-sourcing both the dataset and the code, we invite researchers and practitioners to evaluate any future multilingual models on our benchmark. We provide baselines for five popular multilingual encoder-based language models of different sizes. We hope that this benchmark will foster the creation of novel multilingual legal models and therefore contribute to the progress of natural legal language processing. We imagine this work as a living benchmark and invite the community to extend it with new suitable datasets.

**Future Work** In future work, we plan to extend this benchmark with other NLU tasks and also generation tasks such as summarization, simplification, or translation. Additionally, a deeper analysis of the differences in the behavior of monolingual general-purpose models versus models trained on legal data could provide useful insights for the development of new models. Another relevant aspect that deserves further studies is the impact of the jurisdiction and whether the jurisdiction information is predominantly learned as part of the LLM or is instead learned during fine-tuning. Finally, extending datasets in more languages and evaluating other models such as mT5 (Xue et al., 2021) can be other promising directions.

## Acknowledgements

Joel Niklaus is funded by the Swiss National Research Programme "Digital Transformation" (NRP-77) grant number 187477. Pooja Rani is funded by the Swiss National Science Foundation with Projects No. 200021_197227. Andrea Galassi is funded by the European Commission's NextGeneration EU Programme, PNRR - M4C2 - Investimento 1.3, Partenariato Esteso PE00000013 - FAIR - Future Artificial Intelligence Research - Spoke 8 Pervasive AI. Ilias Chalkidis is funded by the Novo Nordisk Foundation (grant NNF 20SA0066568).

## Limitations

It is important to not exceed the enthusiasm for language models and the ambitions of benchmarks: many recent works have addressed the limits of these tools and analyzed the consequences of their misuse. For example, Bender and Koller (2020) argue that language models do not really learn "meaning". Bender et al. (2021) further expand the discussion by addressing the risks related to these technologies and proposing mitigation methods. Koch et al. (2021) evaluate the use of datasets inside scientific communities and highlight that many machine learning sub-communities focus on very few datasets and that often these dataset are "borrowed" from other communities. Raji et al. (2021) offer a detailed exploration of the limits of popular "general" benchmarks, such as GLUE (Wang et al., 2019b) and ImageNET (Deng et al., 2009). Their analysis covers 3 aspects: limited task design, de-contextualized data and performance reporting, inappropriate community use.

The first problem concerns the fact that typically tasks are not chosen considering proper theories and selecting what would be needed to prove generality. Instead, they are limited to what is considered interesting by the community, what is available, or other similar criteria. These considerations hold also for our work. Therefore, we cannot claim that our benchmark can be used to assess the "generality" of a model or proving that it "understands natural legal language".

The second point addresses the fact that any task, data, or metric are limited to their context, therefore "data benchmarks are closed and inherently subjective, localized constructions". In particular, the content of the data can be too different from real data and the format of the tasks can be too homogeneous compared to human activities. Moreover, any dataset inherently contains biases. We tackle this limitation by deciding to include only tasks and data that are based on real world scenarios, in an effort to minimize the difference between the performance of a model on our benchmark and its performance on a real world problem.

The last aspect regards the negative consequences that benchmarks can have. The competitive testing may encourage misbehavior and the aggregated performance evaluation does create a mirage of cross-domain comparability. The presence of popular benchmarks can influence a scientific community up to the point of steering towards techniques that perform well on that specific benchmark, in disfavor of those that do not. Finally, benchmarks can be misused in marketing to promote commercial products while hiding their flaws. These behaviours obviously cannot be forecasted in advance, but we hope that this analysis of

the shortcomings of our work will be sufficient to prevent misuses of our benchmark and will also inspire research directions for complementary future works. For what specifically concerns aggregated evaluations, they provide an intuitive but imprecise understanding of the performance of a model. While we do not deny their potential downsides, we believe that their responsible use is beneficial, especially when compared to the evaluation of a model on only an arbitrarily selected set of datasets. Therefore, we opted to provide an aggregated evaluation and to weigh languages and tasks equally to make it as robust and fair as possible.

While Raji et al. and Koch et al. argument against the misrepresentations and the misuses of benchmarks and datasets, they do not argue against their usefulness. On the contrary, they consider the creation and adoption of novel benchmarks a sign of a healthy scientific community.

Finally, we want to remark that for many datasets the task of outcome prediction is based not on the document provided by the parties, but on the document provided by the judge along with its decision. For example, Semo et al. (2022) provide a more realistic setup of judgment prediction than other datasets, using actual complaints as inputs. However, due to very limited access to the complaint documents, especially multilingually, creating such datasets is extremely challenging. Thus, most recent works used text from court decisions as proxies. However, predicting the judgment outcome based on text written by the court itself can still be a hard task (as evidenced by results on these datasets). Moreover, it may still require legal reasoning capabilities from models because of the need to pick out the correct information. Additionally, we believe that these tasks can also be interesting to conduct post hoc analyses of decisions.

## Ethics Statement

The scope of this work is to release a unified multilingual legal NLP benchmark to accelerate the development and evaluation of multilingual legal language models. A transparent multilingual and multinational benchmark for NLP in the legal domain might serve as an orientation for scholars and industry researchers by broadening the discussion and helping practitioners to build assisting technology for legal professionals and laypersons. We believe that this is an important application field, where research should be conducted (Tsarapatsanis and Aletras, 2021) to improve legal services and democratize law, while also highlight (inform the audience on) the various multi-aspect shortcomings seeking a responsible and ethical (fair) deployment of legal-oriented technologies.

Nonetheless, irresponsible use (deployment) of such technology is a plausible risk, as in any other application (e.g., online content moderation) and domain (e.g., medical). We believe that similar technologies should only be deployed to assist human experts (e.g., legal scholars in research, or legal professionals in forecasting or assessing legal case complexity) with notices on their limitations.

All datasets included in LEXTREME, are publicly available and have been previously published. We referenced the original work and encourage LEXTREME users to do so as well. In fact, we believe this work should only be referenced, in addition to citing the original work, when experimenting with multiple LEXTREME datasets and using the LEXTREME evaluation infrastructure. Otherwise, only the original work should be cited.

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

## A  Experiment Details

### A.1  Maximum Sequence Lengths

Brazilian Court Decisions: 1024 (128 x 8)
CoVID19: 256
German Argument Mining: 256
Greek Legal Code: 4096 (if speed is important: 2048) (128 x 32 / 16)
Greek Legal NER: 512 (max for non-hierarchical)
LegalNERo: 512 (max for non-hierarchical)
LeNER: 512 (max for non-hierarchical)
MAPA: 512 (max for non-hierarchical)
MultiEURLEX: 4096 (or for maximum performance 8192) (128 x 32 / 64) Online Terms of Service: 256
Swiss Judgment Prediction: 2048 (or for maximum performance on fr: 4096) (128 x 16 / 32)

### A.2  Total compute

We used a total of 689 GPU days.

### A.3  Hyperparameters

We used learning rate 1e-5 for all models and datasets without tuning. We ran all experiments with 3 random seeds (1-3). We always used batch size 64. In case the GPU memory was insufficient, we additionally used gradient accumulation. We trained using early stopping on the validation loss with an early-stopping patience of 5 epochs. Because MultiEURLEX is very large and the experiment very long, we just train for 1 epoch and evaluated after every $1000^{\text{th}}$ step when finetuning multilingual models on the entire dataset. For finetuning the monolingual models on language-specific subsets of MultiEURLEX, we evaluated on the basis of epochs. We used AMP mixed precision training and evaluation to reduce costs. Mixed precision was not used in combination with microsoft/mdeberta-v3-base because it led to errors. For the experiments we used the following NVIDIA GPUs: 24GB RTX3090, 32GB V100 and 80GB A100.

## B  Model Descriptions

**MiniLM.**  MiniLM (Wang et al., 2020) is the result of a novel task-agnostic compression technique, also called distillation, in which a compact model — the so-called student — is trained to reproduce the behaviour of a larger pre-trained model — the so-called teacher. This is achieved by deep self-attention distillation, i.e. only the self-attention module of the last Transformer layer of the teacher, which stores a lot of contextual information (Jawahar et al., 2019), is distilled. The student is trained by closely imitating the teacher's final Transformer layer's self-attention behavior. To aid the learner in developing a better imitation, (Wang et al., 2020) also introduce the self-attention value-relation transfer in addition to the self-attention distributions. The addition of a teacher assistant results in further improvements. For the training of multilingual MiniLM, XLM-$R_{\text{BASE}}$ was used.

**DistilBERT**  DistilBERT (Sanh et al., 2019) is a more compressed version of BERT (Devlin et al., 2019) using teacher-student learning, similar to MiniLM. DistilBERT is distilled from BERT, thus both share a similar overall architecture. The pooler and token-type embeddings are eliminated, and the number of layers is decreased by a factor of 2 in DistilBERT. DistilBERT is distilled in very large batches while utilizing gradient accumulation and dynamic masking, but without the next sentence prediction objective. DistilBERT was trained on the same corpus as the original BERT.

**mDEBERTa**  He et al. (2020) suggest a new model architecture called DeBERTa (Decoding-enhanced BERT with disentangled attention), which employs two novel methods to improve the BERT and RoBERTa models. The first is the disentangled attention mechanism, in which each word is represented by two vectors that encode its content and position, respectively, and the attention weights between words are calculated using disentangled matrices on their respective contents and relative positions. To predict the masked tokens during pre-training, an enhanced mask decoder is utilized, which incorporates absolute positions in the decoding layer. Additionally, the generalization of models is enhanced through fine-tuning using a new virtual adversarial training technique. He et al. (2021) introduce mDEBERTa-v3 by further improving the efficiency of pre-training by

replacing Masked-Language Modeling (MLM) in DeBERTa with the task of replaced token detection (RTD) where the model is trained to predict whether a token in the corrupted input is either original or replaced by agenerator. Further improvements are achieved via *gradient-disentangled embedding sharing* (GDES).

**XLM-RoBERTa** XLM-R (Conneau et al., 2020) is a multilingual language model which has the same pretraining objectives as RoBERTa (Liu et al., 2019), such as dynamic masking, but not next sentence prediction. It is pre-trained on a large corpus comprising 100 languages. The authors report a significant performance gain over multilingual BERT (mBERT) in a variety of tasks with results competitive with state-of-the-art monolingual models (Conneau et al., 2020).

## C  Monolingual Models Overview

| Model | Language | Source | Params | Vocab | Specs |
|---|---|---|---|---|---|
| general | | | | | |
| iarfmoose/roberta-base-bulgarian | bg | - | 126M | 52K | 200K steps / BS 8 |
| UWB-AIR/Czert-B-base-cased | cs | (Sido et al., 2021) | 109M | 31K | 50K steps |
| Maltehb/danish-bert-botxo | da | (Hvingelby et al., 2020) | 111M | 32K | 1M steps / BS 1280 |
| dbmdz/bert-base-german-cased | de | - | 110M | 31K | 1.5M steps |
| deepset/gbert-base | de | (Chan et al., 2020) | 110M | 31K | 30K steps / BS 1024 |
| nlpaueb/bert-base-greek-uncased-v1 | el | (Koutsikakis et al., 2020) | 113M | 35K | 1M steps / BS 256 |
| roberta-base | en | (Liu et al., 2019) | 125M | 50K | 500K steps / BS 8K |
| bertin-project/bertin-roberta-base-spanish | es | (de la Rosa et al., 2022) | 125M | 50K | 250K steps / BS 2048 |
| PlanTL-GOB-ES/roberta-base-bne | es | (Gutiérrez-Fandiño et al., 2021b) | 125M | 50K | 10K steps / BS 2048 |
| tartuNLP/EstBERT | et | (Tanvir et al., 2021) | 124M | 50K | 600K steps / BS 16 |
| TurkuNLP/bert-base-finnish-cased-v1 | fi | (Virtanen et al., 2019) | 125M | 50K | 1M steps / BS 1120 |
| camembert-base | fr | (Martin et al., 2020) | 111M | 32K | 100K steps / BS 8192 |
| dbmdz/bert-base-french-europeana-cased | fr | - | 111M | 32K | 3M steps / BS 128 |
| DCU-NLP/bert-base-irish-cased-v1 | ga | (Barry et al., 2021) | 109M | 30K | 100K steps / BS 128 |
| SZTAKI-HLT/hubert-base-cc | hu | (Nemeskey, 2020) | 111M | 32K | 600K steps / BS 384 |
| Musixmatch/umberto-commoncrawl-cased-v1 | it | - | 111M | 32K | - |
| dbmdz/bert-base-italian-cased | it | - | 110M | 31K | 2-3M steps |
| GroNLP/bert-base-dutch-cased | nl | (de Vries et al., 2019) | 109M | 30K | 850K steps |
| pdelobelle/robbert-v2-dutch-base | nl | (Delobelle et al., 2020) | 117M | 40K | 16K / BS 8192 |
| dkleczek/bert-base-polish-uncased-v1 | pl | | 132M | 60K | 100K steps / BS 256 |
| neuralmind/bert-base-portuguese-cased | pt | (Souza et al., 2020) | 109M | 30K | 1M steps |
| dumitrescustefan/bert-base-romanian-uncased-v1 | ro | (Dumitrescu et al., 2020) | 124M | 50K | 100K steps / BS 20 |
| gerulata/slovakbert | sk | (Pikuliak et al., 2021) | 125M | 50K | 300K steps / BS 512 |
| KB/bert-base-swedish-cased | sv | (Malmsten et al., 2020) | 125M | 50K | 100K steps / BS 128 |
| legal | | | | | |
| zlucia/custom-legalbert | en | (Zheng et al., 2021) | 111M | 32K | 2M steps |
| nlpaueb/legal-bert-base-uncased | en | (Chalkidis et al., 2020) | 109M | 31K | 1M steps / BS 256 |
| PlanTL-GOB-ES/RoBERTalex | es | (Gutiérrez-Fandiño et al., 2021a) | 126M | 52K | BS 2048 |
| dlicari/Italian-Legal-BERT | it | (Licari and Comandé, 2022) | 111M | 32K | 8.4M steps / BS 10 |
| readerbench/jurBERT-base | ro | (Masala et al., 2021) | 111M | 33K | - |

Table 8: Monolingual models. BS is short for batch size. For a detailed overview of the pretraining corpora, we refer to the publications. For some models we were not able to find publications/specs.

.

# D   Dataset Splits

| Language | SJP | OTS-UL | OTS-CT | C19 | MEU-1 | MEU-2 | MEU-3 | MAP-C | MAP-F |
|---|---|---|---|---|---|---|---|---|---|
| bg | | | | | 15986 / 5000 / 5000 | 15986 / 5000 / 5000 | 15986 / 5000 / 5000 | 1411 / 166 / 560 | 1411 / 166 / 560 |
| cs | | | | | 23187 / 5000 / 5000 | 23187 / 5000 / 5000 | 23187 / 5000 / 5000 | 1464 / 176 / 563 | 1464 / 176 / 563 |
| da | | | | | 55000 / 5000 / 5000 | 55000 / 5000 / 5000 | 55000 / 5000 / 5000 | 1455 / 164 / 550 | 1455 / 164 / 550 |
| de | 35458 / 4705 / 9725 | 491 / 42 / 103 | 4480 / 404 / 1027 | | 55000 / 5000 / 5000 | 55000 / 5000 / 5000 | 55000 / 5000 / 5000 | 1457 / 166 / 558 | 1457 / 166 / 558 |
| el | | | | | 55000 / 5000 / 5000 | 55000 / 5000 / 5000 | 55000 / 5000 / 5000 | 1529 / 174 / 584 | 1529 / 174 / 584 |
| en | | 526 / 49 / 103 | 5378 / 415 / 1038 | 648 / 81 / 81 | 55000 / 5000 / 5000 | 55000 / 5000 / 5000 | 55000 / 5000 / 5000 | 893 / 98 / 408 | 893 / 98 / 408 |
| es | | | | | 52785 / 5000 / 5000 | 52785 / 5000 / 5000 | 52785 / 5000 / 5000 | 806 / 248 / 155 | 806 / 248 / 155 |
| et | | | | | 23126 / 5000 / 5000 | 23126 / 5000 / 5000 | 23126 / 5000 / 5000 | 1391 / 163 / 516 | 1391 / 163 / 516 |
| fi | | | | | 42497 / 5000 / 5000 | 42497 / 5000 / 5000 | 42497 / 5000 / 5000 | 1398 / 187 / 531 | 1398 / 187 / 531 |
| fr | 21179 / 3095 / 6820 | | | 1416 / 178 / 178 | 55000 / 5000 / 5000 | 55000 / 5000 / 5000 | 55000 / 5000 / 5000 | 1297 / 97 / 490 | 1297 / 97 / 490 |
| ga | | | | | | | | 1383 / 165 / 515 | 1383 / 165 / 515 |
| hr | | | | 75 / 10 / 10 | 7944 / 2500 / 5000 | 7944 / 2500 / 5000 | 7944 / 2500 / 5000 | | |
| hu | | | | | 22664 / 5000 / 5000 | 22664 / 5000 / 5000 | 22664 / 5000 / 5000 | 1390 / 171 / 525 | 1390 / 171 / 525 |
| it | 3072 / 408 / 812 | 517 / 50 / 102 | 4806 / 432 / 1057 | 742 / 93 / 93 | 55000 / 5000 / 5000 | 55000 / 5000 / 5000 | 55000 / 5000 / 5000 | 1411 / 162 / 550 | 1411 / 162 / 550 |
| lt | | | | | 23188 / 5000 / 5000 | 23188 / 5000 / 5000 | 23188 / 5000 / 5000 | 1413 / 173 / 548 | 1413 / 173 / 548 |
| lv | | | | | 23208 / 5000 / 5000 | 23208 / 5000 / 5000 | 23208 / 5000 / 5000 | 1383 / 167 / 553 | 1383 / 167 / 553 |
| mt | | | | | 17521 / 5000 / 5000 | 17521 / 5000 / 5000 | 17521 / 5000 / 5000 | 937 / 93 / 442 | 937 / 93 / 442 |
| nb | | | | 221 / 28 / 28 | | | | | |
| nl | | | | 135 / 18 / 18 | 55000 / 5000 / 5000 | 55000 / 5000 / 5000 | 55000 / 5000 / 5000 | 1391 / 164 / 530 | 1391 / 164 / 530 |
| pl | | 540 / 50 / 109 | 5278 / 439 / 1175 | 75 / 10 / 10 | 23197 / 5000 / 5000 | 23197 / 5000 / 5000 | 23197 / 5000 / 5000 | | |
| pt | | | | | 52370 / 5000 / 5000 | 52370 / 5000 / 5000 | 52370 / 5000 / 5000 | 1086 / 105 / 390 | 1086 / 105 / 390 |
| ro | | | | | 15921 / 5000 / 5000 | 15921 / 5000 / 5000 | 15921 / 5000 / 5000 | 1480 / 175 / 557 | 1480 / 175 / 557 |
| sk | | | | | 22971 / 5000 / 5000 | 22971 / 5000 / 5000 | 22971 / 5000 / 5000 | 1395 / 165 / 526 | 1395 / 165 / 526 |
| sl | | | | | 23184 / 5000 / 5000 | 23184 / 5000 / 5000 | 23184 / 5000 / 5000 | | |
| sv | | | | | 42490 / 5000 / 5000 | 42490 / 5000 / 5000 | 42490 / 5000 / 5000 | 1453 / 175 / 539 | 1453 / 175 / 539 |

Table 9: Overview of the number of examples for each language-specific subset of multilingual tasks. The order of the values is train / validation / test.

| Language | SJP | OTS-UL | OTS-CT | C19 | MEU-1 | MEU-2 | MEU-3 | MAP-C | MAP-F |
|---|---|---|---|---|---|---|---|---|---|
| bg | | | | | 21 / 21 / 21 | 127 / 126 / 127 | 481 / 454 / 465 | 11 / 11 / 8 | 24 / 16 / 13 |
| cs | | | | | 21 / 21 / 21 | 127 / 126 / 127 | 486 / 454 / 465 | 11 / 11 / 9 | 30 / 17 / 16 |
| da | | | | | 21 / 21 / 21 | 127 / 126 / 127 | 500 / 454 / 465 | 11 / 10 / 11 | 26 / 14 / 14 |
| de | 2 / 2 / 2 | 3 / 3 / 3 | 9 / 7 / 9 | | 21 / 21 / 21 | 127 / 126 / 127 | 500 / 454 / 465 | 11 / 9 / 10 | 28 / 14 / 14 |
| el | | | | | 21 / 21 / 21 | 127 / 126 / 127 | 500 / 454 / 465 | 11 / 11 / 11 | 31 / 17 / 20 |
| en | | 3 / 3 / 3 | 9 / 8 / 9 | 6 / 6 / 5 | 21 / 21 / 21 | 127 / 126 / 127 | 500 / 454 / 465 | 11 / 9 / 9 | 28 / 17 / 18 |
| es | | | | | 21 / 21 / 21 | 127 / 126 / 127 | 497 / 454 / 465 | 11 / 8 / 11 | 26 / 13 / 18 |
| et | | | | | 21 / 21 / 21 | 127 / 126 / 127 | 486 / 454 / 465 | 11 / 11 / 11 | 25 / 14 / 17 |
| fi | | | | | 21 / 21 / 21 | 127 / 126 / 127 | 493 / 454 / 465 | 11 / 11 / 10 | 24 / 19 / 16 |
| fr | 2 / 2 / 2 | | | 8 / 8 / 7 | 21 / 21 / 21 | 127 / 126 / 127 | 500 / 454 / 465 | 11 / 11 / 11 | 32 / 19 / 26 |
| ga | | | | | | | | 13 / 11 / 11 | 33 / 17 / 18 |
| hr | | | | | 21 / 21 / 21 | 127 / 126 / 127 | 469 / 437 / 465 | | |
| hu | | | | 4 / 1 / 1 | 21 / 21 / 21 | 127 / 126 / 127 | 486 / 454 / 465 | 11 / 10 / 10 | 20 / 15 / 14 |
| it | 2 / 2 / 2 | 3 / 3 / 3 | 9 / 8 / 9 | 7 / 7 / 6 | 21 / 21 / 21 | 127 / 126 / 127 | 500 / 454 / 465 | 11 / 10 / 11 | 25 / 15 / 16 |
| lt | | | | | 21 / 21 / 21 | 127 / 126 / 127 | 486 / 454 / 465 | 11 / 11 / 10 | 28 / 19 / 21 |
| lv | | | | | 21 / 21 / 21 | 127 / 126 / 127 | 486 / 454 / 465 | 11 / 11 / 11 | 31 / 15 / 21 |
| mt | | | | | 21 / 21 / 21 | 127 / 126 / 127 | 485 / 454 / 465 | 11 / 11 / 11 | 27 / 15 / 15 |
| nb | | | | 7 / 5 / 6 | | | | | |
| nl | | | | 2 / 2 / 2 | 21 / 21 / 21 | 127 / 126 / 127 | 500 / 454 / 465 | 10 / 9 / 10 | 25 / 12 / 14 |
| pl | | 3 / 3 / 3 | 9 / 8 / 9 | 7 / 5 / 3 | 21 / 21 / 21 | 127 / 126 / 127 | 486 / 454 / 465 | | |
| pt | | | | | 21 / 21 / 21 | 127 / 126 / 127 | 497 / 454 / 465 | 11 / 10 / 11 | 29 / 14 / 18 |
| ro | | | | | 21 / 21 / 21 | 127 / 126 / 127 | 481 / 454 / 465 | 11 / 11 / 11 | 25 / 16 / 18 |
| sk | | | | | 21 / 21 / 21 | 127 / 126 / 127 | 485 / 454 / 465 | 11 / 11 / 11 | 25 / 16 / 18 |
| sl | | | | | 21 / 21 / 21 | 127 / 126 / 127 | 486 / 454 / 465 | | |
| sv | | | | | 21 / 21 / 21 | 127 / 126 / 127 | 493 / 454 / 465 | 11 / 11 / 10 | 23 / 15 / 15 |

Table 10: Overview of the number of labels for each language-specific subset of multilingual tasks. The order of the values is train / validation / test.

# E   Detailed Multilingual Results

| Model | Mean | BCD-J | BCD-U | GAM | GLC-V | GLC-C | GLC-S | SJP | OTS-UL | OTS-CT | C19 | MEU-1 | MEU-2 | MEU-3 | GLN | LNR | LNB | MAP-C | MAP-F |
|---|---|---|---|---|---|---|---|---|---|---|---|---|---|---|---|---|---|---|---|
| MiniLM | 51.0 | 52.8 (±6.7) | 55.1 (±6.6) | 72.1 (±0.9) | 82.0 (±1.0) | 39.4 (±1.0) | **5.1 (±1.6)** | 68.9 (±0.7) | 71.0 (±5.0) | 15.3 (±3.4) | 5.8 (±1.5) | 64.8 (±0.3) | 23.5 (±0.6) | 6.5 (±0.3) | **66.6 (±1.4)** | 62.9 (±6.4) | **98.2 (±0.1)** | **79.0 (±0.3)** | 49.3 (±1.4) |
| DistilBERT | 59.2 | 52.1 (±4.5) | 60.0 (±9.8) | 70.6 (±1.7) | 84.9 (±0.5) | 68.0 (±0.6) | 33.9 (±2.0) | 68.7 (±0.7) | 66.9 (±3.4) | 49.6 (±9.1) | 41.4 (±5.6) | 68.2 (±0.1) | 37.3 (±0.3) | 13.8 (±0.6) | 62.4 (±4.5) | 69.9 (±8.9) | 93.0 (±2.6) | 76.3 (±1.2) | 48.6 (±3.2) |
| mDeBERTa v3 | 60.1 | **70.4 (±0.6)** | 67.5 (±6.1) | 70.0 (±1.1) | 85.0 (±0.8) | 58.2 (±7.5) | 12.3 (±2.5) | **71.2 (±0.7)** | **85.2 (±2.9)** | 52.1 (±4.6) | 43.4 (±4.3) | 68.4 (±0.6) | 36.4 (±0.9) | 14.0 (±1.3) | 63.7 (±4.6) | 63.9 (±8.5) | 96.2 (±1.7) | 74.3 (±2.9) | 49.8 (±2.4) |
| XLM-R$_{Base}$ | 59.0 | 67.5 (±2.2) | 63.4 (±12.3) | 72.5 (±1.9) | **85.6 (±0.2)** | 69.1 (±0.6) | 15.7 (±12.7) | 69.6 (±0.9) | 72.6 (±4.2) | 52.4 (±6.0) | 44.1 (±7.9) | 69.2 (±0.1) | 32.2 (±1.7) | 5.4 (±0.5) | 64.2 (±1.7) | 57.0 (±3.5) | 96.4 (±0.9) | 73.4 (±1.9) | 51.1 (±2.7) |
| XLM-R$_{Large}$ | **63.6** | 58.1 (±9.3) | **70.4 (±3.7)** | **73.0 (±1.4)** | 58.2 (±50.2) | **73.0 (±0.9)** | 38.9 (±33.7) | 70.0 (±1.8) | 84.9 (±2.7) | **62.9 (±6.1)** | **53.8 (±10.5)** | **71.2 (±1.4)** | **47.6 (±0.4)** | **15.3 (±0.8)** | 63.0 (±4.0) | **75.2 (±3.0)** | 96.6 (±1.1) | 77.1 (±2.0) | **55.8 (±3.6)** |

Table 11: Arithmetic mean of macro-F1 and the standard deviation over all seeds for multilingual models from the validation set. The best scores are in bold.

| Model | Mean | BCD-J | BCD-U | GAM | GLC-V | GLC-C | GLC-S | SJP | OTS-UL | OTS-CT | C19 | MEU-1 | MEU-2 | MEU-3 | GLN | LNR | LNB | MAP-C | MAP-F |
|---|---|---|---|---|---|---|---|---|---|---|---|---|---|---|---|---|---|---|---|
| MiniLM | 46.4 | 49.4 (±7.4) | 56.7 (±7.9) | **73.3 (±0.9)** | 81.7 (±0.5) | 39.4 (±1.4) | **5.2 (±1.6)** | 67.6 (±1.2) | 74.6 (±1.1) | 14.1 (±3.1) | **6.0 (±1.9)** | 62.0 (±0.4) | 21.7 (±0.5) | 5.6 (±0.3) | 43.6 (±2.4) | 46.5 (±1.5) | 86.1 (±0.3) | 62.7 (±1.7) | 39.9 (±2.3) |
| DistilBERT | 53.5 | 50.3 (±2.9) | 58.8 (±8.7) | 69.5 (±0.9) | 85.2 (±0.8) | 70.0 (±0.3) | 33.2 (±1.9) | 66.7 (±1.1) | 67.2 (±4.1) | 46.2 (±8.9) | 39.5 (±6.3) | 63.6 (±0.1) | 33.6 (±0.5) | 12.0 (±0.8) | 38.1 (±2.0) | 48.4 (±5.2) | 78.7 (±1.1) | 61.3 (±2.8) | 40.6 (±0.7) |
| mDeBERTa v3 | 55.2 | **67.2 (±1.9)** | 53.2 (±6.7) | 71.3 (±0.3) | 85.6 (±1.0) | 58.6 (±7.8) | 12.4 (±2.8) | **69.0 (±0.8)** | 79.7 (±3.8) | 53.8 (±3.0) | 40.7 (±5.0) | 65.0 (±0.4) | 34.1 (±1.0) | 13.1 (±1.0) | 45.1 (±3.9) | 46.7 (±0.7) | 87.3 (±1.1) | **65.6 (±4.7)** | **45.7 (±1.0)** |
| XLM-R$_{Base}$ | 55.6 | 65.4 (±3.6) | 61.6 (±11.2) | 72.0 (±2.4) | **86.1 (±0.4)** | 70.7 (±1.0) | 15.4 (±12.3) | 68.3 (±1.0) | 80.8 (±1.9) | 55.9 (±2.6) | 45.9 (±11.0) | 65.6 (±0.1) | 29.8 (±1.5) | 4.7 (±0.4) | **46.4 (±1.9)** | 45.6 (±0.6) | 87.4 (±1.0) | 58.0 (±2.4) | 41.8 (±2.4) |
| XLM-R$_{Large}$ | **58.9** | 55.1 (±7.6) | **62.3 (±3.6)** | 73.1 (±1.5) | 58.3 (±50.3) | **74.7 (±0.9)** | 39.1 (±33.9) | 68.3 (±1.8) | **83.6 (±4.8)** | **66.9 (±0.5)** | 54.2 (±7.2) | **68.1 (±1.2)** | **44.4 (±0.3)** | **14.2 (±0.8)** | 45.3 (±3.1) | **55.8 (±5.9)** | **88.4 (±1.2)** | 64.0 (±2.5) | 43.7 (±1.1) |

Table 12: Arithmetic mean of macro-F1 and the standard deviation over all seeds for multilingual models from the test set. The best scores are in bold.

# F   Detailed Monolingual Results

Table 13 (rotated; columns in original order): Model, Mean, BCD-J, BCD-U, GAM, GLC-V, GLC-C, GLC-S, SJP, OTS-UL, OTS-CT, C19, MEU-1, MEU-2, MEU-3, GLN, LNR, LNB, MAP-C, MAP-F.

| Model | Mean | BCD-J | BCD-U | GAM | GLC-V | GLC-C | GLC-S | SJP | OTS-UL | OTS-CT | C19 | MEU-1 | MEU-2 | MEU-3 | GLN | LNR | LNB | MAP-C | MAP-F |
|---|---|---|---|---|---|---|---|---|---|---|---|---|---|---|---|---|---|---|---|
| iarfmoose/roberta-base-bulgarian | 52.3 | - | - | - | - | - | - | - | - | - | - | 67.4 (±0.7) | 35.7 (±2.1) | 3.7 (±5.2) | - | - | - | 82.8 (±2.1) | 72.0 (±2.8) |
| UWB-AIR/Czert-B-base-cased | 61.7 | - | - | - | - | - | - | - | - | - | - | 69.1 (±1.1) | 51.9 (±0.6) | 27.3 (±0.8) | - | - | - | 92.2 (±0.8) | 67.8 (±3.1) |
| Maltehb/danish-bert-botxo | 57.7 | - | - | - | - | - | - | - | - | - | - | 69.5 (±1.7) | 46.7 (±2.7) | 25.8 (±1.0) | - | - | - | 89.2 (±3.5) | 57.3 (±5.4) |
| dbmdz/bert-base-german-cased | 63.2 | - | - | 72.8 (±0.9) | - | - | - | 70.9 (±1.1) | 76.3 (±5.4) | 46.2 (±1.9) | - | 71.8 (±1.3) | 51.1 (±1.4) | 27.2 (±0.1) | - | - | - | 85.5 (±8.6) | 67.3 (±4.0) |
| deepset/gbert-base | 62.3 | - | - | 74.6 (±1.0) | - | - | - | 71.5 (±1.2) | 89.0 (±7.1) | 50.4 (±1.2) | - | 71.6 (±1.3) | 49.9 (±0.2) | 25.9 (±0.5) | - | - | - | 74.4 (±6.6) | 53.2 (±9.2) |
| nlpaueb/bert-base-greek-uncased-v1 | 66.1 | - | - | - | 87.9 (±0.4) | 74.5 (±0.3) | 61.7 (±0.3) | - | - | - | - | 71.0 (±0.4) | 48.6 (±0.1) | 25.9 (±0.2) | 65.4 (±3.2) | - | - | 98.2 (±1.3) | 61.3 (±1.4) |
| roberta-base | 44.2 | - | - | - | - | - | - | - | 92.3 (±0.3) | 65.6 (±6.9) | 35.0 (±2.5) | 48.5 (±1.0) | 0.0 (±0.0) | 0.0 (±0.0) | - | - | - | 75.4 (±2.2) | 37.2 (±6.7) |
| bertin-project/bertin-roberta-base-spanish | 44.4 | - | - | - | - | - | - | - | - | - | - | 66.7 (±1.2) | 33.4 (±2.7) | 17.4 (±2.1) | - | - | - | 46.8 (±14.6) | 57.6 (±5.5) |
| PlanTL-GOB-ES/roberta-base-bne | 52.4 | - | - | - | - | - | - | - | - | - | - | 67.5 (±0.4) | 32.6 (±2.4) | 15.1 (±0.9) | - | - | - | 77.7 (±14.3) | 69.3 (±12.5) |
| tartuNLP/EstBERT | 58.3 | - | - | - | - | - | - | - | - | - | - | 67.7 (±1.5) | 47.7 (±0.4) | 24.6 (±0.7) | - | - | - | 90.9 (±2.0) | 60.5 (±7.1) |
| TurkuNLP/bert-base-finnish-cased-v1 | 61.8 | - | - | - | - | - | - | - | - | - | - | 70.4 (±1.4) | 49.6 (±1.0) | 27.7 (±1.2) | - | - | - | 94.2 (±4.0) | 67.2 (±6.6) |
| camembert-base | 30.8 | - | - | - | - | - | - | 68.8 (±1.3) | - | - | 16.7 (±1.7) | 69.6 (±0.2) | 41.9 (±3.6) | 18.6 (±2.9) | - | - | - | 0.0 (±0.0) | 0.0 (±0.0) |
| dbmdz/bert-base-french-europeana-cased | 57.1 | - | - | - | - | - | - | 70.2 (±0.6) | - | - | 36.2 (±5.6) | 70.1 (±1.2) | 48.9 (±0.0) | 25.7 (±0.7) | - | - | - | 84.5 (±6.6) | 63.9 (±5.6) |
| DCU-NLP/bert-base-irish-cased-v1 | 72.3 | - | - | - | - | - | - | - | - | - | - | - | - | - | - | - | - | 79.6 (±0.4) | 65.1 (±3.5) |
| SZTAKI-HLT/hubert-base-cc | 50.6 | - | - | - | - | - | - | - | - | - | 0.0 (±0.0) | 69.5 (±2.5) | 53.1 (±1.6) | 27.9 (±2.1) | - | - | - | 91.2 (±1.5) | 62.0 (±1.3) |
| Musixmatch/umberto-commoncrawl-cased-v1 | 38.5 | - | - | - | - | - | - | 57.1 (±5.9) | 90.7 (±7.3) | 42.1 (±4.8) | 13.5 (±0.9) | 69.5 (±0.9) | 43.5 (±1.5) | 21.7 (±0.7) | - | - | - | 8.0 (±7.3) | 0.0 (±0.0) |
| dbmdz/bert-base-italian-cased | 53.6 | - | - | - | - | - | - | 56.5 (±3.6) | 57.2 (±1.1) | 51.5 (±2.8) | 31.5 (±2.1) | 70.9 (±1.3) | 48.5 (±0.5) | 25.4 (±0.4) | - | - | - | 87.5 (±3.8) | 53.2 (±3.8) |
| GroNLP/bert-base-dutch-cased | 48.7 | - | - | - | - | - | - | - | - | - | 2.1 (±1.8) | 70.1 (±1.3) | 51.8 (±0.6) | 26.5 (±0.6) | - | - | - | 86.2 (±11.3) | 55.5 (±5.5) |
| pdelobelle/robbert-v2-dutch-base | 42.1 | - | - | - | - | - | - | - | - | - | 0.0 (±0.0) | 69.7 (±0.4) | 42.1 (±3.0) | 16.7 (±0.0) | - | - | - | 77.9 (±0.8) | 46.0 (±1.1) |
| dkleczek/bert-base-polish-uncased-v1 | 50.6 | - | - | - | - | - | - | - | 73.0 (±19.0) | 54.9 (±1.1) | 29.2 (±3.6) | 69.9 (±1.8) | 50.0 (±2.8) | 26.4 (±0) | - | - | - | - | - |
| neuralmind/bert-base-portuguese-cased | 65.8 | 66.7 (±4.1) | 71.2 (±3.6) | - | - | - | - | - | - | - | - | 68.7 (±3.9) | 48.4 (±3.9) | 26.2 (±1.1) | - | - | 96.6 (±0.0) | 96.2 (±3.5) | 52.7 (±8.3) |
| dumitrescustefan/bert-base-romanian-uncased-v1 | 59.9 | - | - | - | - | - | - | - | - | - | - | 70.0 (±0.4) | 51.3 (±0.6) | 26.8 (±0) | - | 55.1 (±0.7) | - | 86.8 (±2.1) | 69.5 (±1.4) |
| gerulata/slovakbert | 47.5 | - | - | - | - | - | - | - | - | - | - | 70.1 (±0.9) | 41.7 (±2.0) | 13.0 (±0.6) | - | - | - | 74.3 (±1.6) | 38.2 (±4.3) |
| KB/bert-base-swedish-cased | 63.1 | - | - | - | - | - | - | - | - | - | - | 70.8 (±1.4) | 51.8 (±0.3) | 27.8 (±0.8) | - | - | - | 95.8 (±1.6) | 69.5 (±4.3) |
| zlucia/custom-legalbert | 55.0 | - | - | - | - | - | - | - | 77.8 (±16.9) | 68.1 (±3.5) | 33.1 (±1.9) | 68.6 (±2.0) | 45.7 (±5.3) | 24.5 (±0.4) | - | - | - | 74.3 (±2.0) | 48.3 (±10.6) |
| nlpaueb/legal-bert-base-uncased | 56.3 | - | - | - | - | - | - | - | 93.4 (±4.1) | 67.8 (±3.6) | 29.4 (±3.0) | 71.2 (±0.5) | 50.0 (±1.9) | 25.9 (±0.5) | - | - | - | 78.0 (±0.5) | 35.0 (±7.6) |
| PlanTL-GOB-ES/RoBERTalex | 52.8 | - | - | - | - | - | - | - | - | - | - | 69.7 (±1.1) | 45.5 (±1.6) | 21.9 (±0) | - | - | - | 75.8 (±1.6) | 51.3 (±2.4) |
| dlicari/Italian-Legal-BERT | 45.4 | - | - | - | - | - | - | 56.8 (±5.9) | 61.7 (±8.5) | 41.2 (±5.9) | 24.6 (±2.8) | 67.0 (±1.2) | 42.6 (±0.6) | 20.3 (±0.7) | - | - | - | 62.8 (±10.4) | 31.9 (±17.5) |
| readerbench/jurBERT-base | 51.9 | - | - | - | - | - | - | - | - | - | - | 69.1 (±1.6) | 48.2 (±2.6) | 23.2 (±0.4) | - | 52.7 (±7.8) | - | 76.8 (±4.0) | 41.2 (±6.7) |

Table 13: Arithmetic mean of macro-F1 and the standard deviation over all seeds for monolingual models from the validation set.

| Model | Mean | BCD-J | BCD-U | GAM | GLC-V | GLC-C | GLC-S | SJP | OTS-UL | OTS-CT | C19 | MEU-1 | MEU-2 | MEU-3 | GLN | LNR | LNB | MAP-C | MAP-F |
|---|---|---|---|---|---|---|---|---|---|---|---|---|---|---|---|---|---|---|---|
| iarfmoose/roberta-base-bulgarian | 46.7 | - | - | - | - | - | - | - | - | - | - | 63.1 (±1.8) | 32.6 (±1.8) | 3.2 (±4.5) | - | - | - | 62.5 (±3.6) | 72.1 (±8.2) |
| UWB-AIR/Czert-B-base-cased | 51.6 | - | - | - | - | - | - | - | - | - | - | 65.4 (±1.0) | 47.9 (±0.1) | 25.2 (±0.4) | - | - | - | 64.8 (±7.8) | 54.9 (±4.2) |
| Maltehb/danish-bert-botxo | 46.7 | - | - | - | - | - | - | - | - | - | - | 65.8 (±1.6) | 43.3 (±2.7) | 24.3 (±0.8) | - | - | - | 54.0 (±0.8) | 46.2 (±5.5) |
| dbmdz/bert-base-german-cased | 56.1 | - | - | 72.6 (±1.4) | - | - | - | 68.7 (±1.0) | 68.8 (±3.5) | 54.9 (±4.0) | - | 68.1 (±0.9) | 47.6 (±1.1) | 25.0 (±0.1) | - | - | - | 57.2 (±4.2) | 41.7 (±2.1) |
| deepset/gbert-base | 57.2 | - | - | 75.1 (±1.3) | - | - | - | 69.3 (±0.7) | 74.2 (±0.4) | 50.8 (±3.8) | - | 67.9 (±0.8) | 45.7 (±0.6) | 23.6 (±0.6) | - | - | - | 58.6 (±1.2) | 49.4 (±2.9) |
| nlpaueb/bert-base-greek-uncased-v1 | 56.0 | - | - | - | 88.1 (±0.6) | 76.5 (±0.7) | 62.8 (±0.4) | - | - | - | - | 67.6 (±0.1) | 45.7 (±0.2) | 23.5 (±0.5) | - | - | - | 54.6 (±4.9) | 38.3 (±0.2) |
| roberta-base | 41.9 | - | - | - | - | - | - | - | 67.2 (±10.3) | 69.5 (±0.7) | 37.0 (±1.9) | 47.5 (±0.9) | 0.0 (±0.0) | 0.0 (±0.0) | 47.0 (±4.3) | - | - | 66.2 (±4.7) | 48.0 (±4.7) |
| bertin-project/bertin-roberta-base-spanish | 41.1 | - | - | - | - | - | - | - | - | - | - | 63.4 (±0.7) | 30.1 (±2.3) | 15.4 (±1.8) | - | - | - | 42.9 (±13.3) | 53.6 (±5.2) |
| PlanTL-GOB-ES/roberta-base-bne | 41.8 | - | - | - | - | - | - | - | - | - | - | 64.3 (±0.5) | 28.8 (±2.2) | 13.1 (±1.0) | - | - | - | 52.3 (±10.2) | 50.3 (±2.5) |
| tartuNLP/EstBERT | 42.9 | - | - | - | - | - | - | - | - | - | - | 64.0 (±1.3) | 43.0 (±0.6) | 21.9 (±0.8) | - | - | - | 36.4 (±1.9) | 49.1 (±6.6) |
| TurkuNLP/bert-base-finnish-cased-v1 | 47.2 | - | - | - | - | - | - | - | - | - | - | 66.8 (±1.0) | 45.7 (±0.3) | 25.1 (±1.5) | - | - | - | 53.3 (±0.4) | 45.0 (±7.7) |
| camembert-base | 29.3 | - | - | - | - | - | - | 69.7 (±1.4) | - | - | 13.7 (±1.2) | 66.2 (±0.4) | 38.4 (±2.9) | 17.4 (±2.6) | - | - | - | 0.0 (±0.0) | 0.0 (±0.0) |
| dbmdz/bert-base-french-europeana-cased | 47.1 | - | - | - | - | - | - | 70.2 (±1.4) | - | - | 36.0 (±1.9) | 65.7 (±1.1) | 45.2 (±0.4) | 23.7 (±0.7) | - | - | - | 52.0 (±3.4) | 36.7 (±3.3) |
| DCU-NLP/bert-base-irish-cased-v1 | 44.1 | - | - | - | - | - | - | - | - | - | - | - | - | - | - | - | - | 42.3 (±1.3) | 46.0 (±0.4) |
| SZTAKI-HLT/hubert-base-cc | 41.2 | - | - | - | - | - | - | - | - | - | 0.0 (±0.0) | 66.1 (±2.1) | 48.1 (±0.4) | 24.7 (±1.6) | - | - | - | 49.0 (±2.9) | 59.6 (±0.4) |
| Musixmatch/umberto-commoncrawl-cased-v1 | 36.1 | - | - | - | - | - | - | 57.4 (±1.1) | 76.9 (±3.2) | 40.9 (±2.8) | 16.3 (±0.5) | 65.9 (±1.1) | 39.9 (±1.5) | 20.0 (±0.7) | - | - | - | 7.7 (±7.0) | 0.0 (±0.0) |
| dbmdz/bert-base-italian-cased | 48.1 | - | - | - | - | - | - | 57.3 (±3.4) | 69.8 (±1.6) | 47.8 (±5.2) | 29.5 (±3.2) | 67.5 (±1.1) | 45.4 (±0.3) | 23.1 (±0.4) | - | - | - | 47.7 (±1.8) | 45.1 (±5.0) |
| GroNLP/bert-base-dutch-cased | 40.0 | - | - | - | - | - | - | - | - | - | 4.4 (±0.5) | 66.1 (±1.2) | 47.2 (±0.4) | 24.0 (±0.2) | - | - | - | 51.2 (±2.3) | 47.2 (±2.0) |
| pdelobelle/robbert-v2-dutch-base | 37.0 | - | - | - | - | - | - | - | - | - | 0.0 (±0.0) | 66.0 (±0.2) | 38.4 (±2.6) | 15.3 (±0.1) | - | - | - | 56.2 (±2.2) | 45.9 (±1.4) |
| dkleczek/bert-base-polish-uncased-v1 | 45.9 | - | - | - | - | - | - | - | 70.6 (±3.0) | 58.6 (±5.7) | 11.3 (±3.6) | 66.0 (±1.6) | 45.5 (±2.3) | 23.4 (±0) | - | - | 87.2 (±0.2) | - | - |
| neuralmind/bert-base-portuguese-cased | 57.6 | 64.5 (±6.3) | 70.6 (±8.2) | - | - | - | - | - | - | - | - | 65.7 (±3.5) | 44.8 (±0.6) | 23.9 (±0.8) | - | - | - | 62.8 (±0.2) | 41.7 (±1.7) |
| dumitrescustefan/bert-base-romanian-uncased-v1 | 48.6 | - | - | - | - | - | - | - | - | - | - | 66.7 (±0.7) | 46.8 (±0.3) | 23.5 (±0) | - | 43.1 (±2.4) | - | 51.5 (±1.3) | 60.1 (±1.2) |
| gerulata/slovakbert | 38.4 | - | - | - | - | - | - | - | - | - | - | 66.6 (±0.4) | 37.5 (±2.2) | 11.9 (±0.4) | - | - | - | 42.0 (±0.9) | 33.8 (±0.5) |
| KB/bert-base-swedish-cased | 50.0 | - | - | - | - | - | - | - | - | - | - | 67.4 (±1.5) | 47.4 (±0.5) | 25.6 (±0.5) | - | - | - | 56.0 (±2.2) | 53.8 (±6.6) |
| zlucia/custom-legalbert | 50.2 | - | - | - | - | - | - | - | 67.5 (±6.7) | 69.1 (±4.3) | 32.7 (±2.7) | 65.0 (±2.1) | 42.2 (±5.3) | 22.6 (±0.4) | - | - | - | 57.3 (±2.8) | 45.4 (±2.9) |
| nlpaueb/legal-bert-base-uncased | 55.3 | - | - | - | - | - | - | - | 88.9 (±3.5) | 71.2 (±2.5) | 29.4 (±4.8) | 67.6 (±0.7) | 46.5 (±1.2) | 24.3 (±0.5) | - | - | - | 67.7 (±2.1) | 46.8 (±6.4) |
| PlanTL-GOB-ES/RoBERTalex | 45.8 | - | - | - | - | - | - | - | - | - | - | 65.9 (±1.1) | 42.3 (±1.6) | 20.1 (±0) | - | - | - | 52.2 (±6.1) | 48.3 (±1.3) |
| dlicari/Italian-Legal-BERT | 43.0 | - | - | - | - | - | - | 60.6 (±7.8) | 76.5 (±2.9) | 32.7 (±3.2) | 22.3 (±2.6) | 63.5 (±0.8) | 39.3 (±0.1) | 18.8 (±0.8) | - | 42.1 (±1.4) | - | 40.3 (±3.4) | 33.0 (±16.1) |
| readerbench/jurBERT-base | 42.2 | - | - | - | - | - | - | - | - | - | - | 65.0 (±0.7) | 43.2 (±1.9) | 20.8 (±0.4) | - | - | - | 43.2 (±1.7) | 39.1 (±5.5) |

Table 14: Arithmetic mean of macro-F1 and the standard deviation over all seeds for monolingual models from the test set.

# G Original Paper Results

In this section, we present an overview of scores for each configuration of the LEXTREME dataset as provided in the original papers. When certain configurations were not available, no scores were obtained. It should be noted that different papers provide varying scores, making direct comparisons with our results challenging. Additionally, the variability in the training and evaluation procedure used across different papers may impact the resulting scores, which is an important factor to consider. To gain a better understanding of the training and evaluation procedure please refer to the cited references. The LEXTREME scores are calculated by taking the arithmetic mean of each seed (three in total).

| Source | Method | TrainLang | TestLang | macro-precision | macro-recall | macro-f1 | micro-precision | micro-recall | micro-f1 | precision | recall | accuracy |
|---|---|---|---|---|---|---|---|---|---|---|---|---|
| Lage-Freitas et al. (2022) | BERT-Imbau | pt | pt | - | - | **73.0** | - | - | - | 66.0 | 63.0 | 73.0 |
| Lage-Freitas et al. (2022) | Bidirectional Long Short-Term Memory (BiLSTM) | pt | pt | - | - | 55.0 | - | - | - | 43.0 | 41.0 | 55.0 |
| Lage-Freitas et al. (2022) | Convolutional Neural Networks (CNN) | pt | pt | - | - | 61.0 | - | - | - | 65.0 | 59.0 | 70.0 |
| Lage-Freitas et al. (2022) | decision tree | pt | pt | - | - | 63.0 | - | - | - | 67.0 | 61.0 | 72.0 |
| Lage-Freitas et al. (2022) | eXtreme Gradient Boosting (XGBoost) | pt | pt | - | - | 70.0 | - | - | - | **73.0** | **68.0** | **77.0** |
| Lage-Freitas et al. (2022) | Gated Recurrent Unit (GRU) | pt | pt | - | - | 72.0 | - | - | - | 66.0 | 61.0 | 72.0 |
| Lage-Freitas et al. (2022) | Gaussian Naive Bayes (GNB) | pt | pt | - | - | 48.0 | - | - | - | 47.0 | 48.0 | 55.0 |
| Lage-Freitas et al. (2022) | Long Short-Term Memory (LSTM) | pt | pt | - | - | 71.0 | - | - | - | 64.0 | 61.0 | 71.0 |
| Lage-Freitas et al. (2022) | random forest | pt | pt | - | - | 29.0 | - | - | - | 52.0 | 36.0 | 61.0 |
| Lage-Freitas et al. (2022) | support vector machine | pt | pt | - | - | 68.0 | - | - | - | 72.0 | 66.0 | 76.0 |
| lextreme | distilbert-base-multilingual-cased | pt | pt | 51.3 | 51.9 | 50.3 | 54.0 | 54.0 | 54.0 | - | - | 54.0 |
| lextreme | microsoft/mdeberta-v3-base | pt | pt | **66.2** | **69.1** | 67.2 | **71.5** | **71.5** | **71.5** | - | - | 71.5 |
| lextreme | microsoft/Multilingual-MiniLM-L12-H384 | pt | pt | 51.9 | 53.8 | 49.4 | 50.2 | 50.2 | 50.2 | - | - | 50.2 |
| lextreme | neuralmind/bert-base-portuguese-cased | pt | pt | 64.5 | 68.5 | 64.5 | 67.2 | 67.2 | 67.2 | - | - | 67.2 |
| lextreme | xlm-roberta-base | pt | pt | 64.7 | 68.3 | 65.4 | 69.1 | 69.1 | 69.1 | - | - | 69.1 |
| lextreme | xlm-roberta-large | pt | pt | 53.3 | 59.2 | 55.1 | 63.2 | 63.2 | 63.2 | - | - | 63.2 |

Table 15: BCD-J. The best scores are in bold.

| Source | Method | TrainLang | TestLang | macro-precision | macro-recall | macro-f1 | micro-precision | micro-recall | micro-f1 | precision | recall | accuracy |
|---|---|---|---|---|---|---|---|---|---|---|---|---|
| Lage-Freitas et al. (2022) | BERT-Imbau | pt | pt | - | - | 98.0 | - | - | - | 59.0 | 53.0 | 98.0 |
| Lage-Freitas et al. (2022) | Bidirectional Long Short-Term Memory (BiLSTM) | pt | pt | - | - | **99.0** | - | - | - | 80.0 | 65.0 | **99.0** |
| Lage-Freitas et al. (2022) | Convolutional Neural Networks (CNN) | pt | pt | - | - | **99.0** | - | - | - | 88.0 | 69.0 | **99.0** |
| Lage-Freitas et al. (2022) | decision tree | pt | pt | - | - | 81.0 | - | - | - | 88.0 | **77.0** | **99.0** |
| Lage-Freitas et al. (2022) | eXtreme Gradient Boosting (XGBoost) | pt | pt | - | - | 81.0 | - | - | - | **92.0** | 74.0 | **99.0** |
| Lage-Freitas et al. (2022) | Gated Recurrent Unit (GRU) | pt | pt | - | - | **99.0** | - | - | - | 84.0 | 65.0 | **99.0** |
| Lage-Freitas et al. (2022) | Gaussian Naive Bayes (GNB) | pt | pt | - | - | 64.0 | - | - | - | 73.0 | 61.0 | 98.0 |
| Lage-Freitas et al. (2022) | Long Short-Term Memory (LSTM) | pt | pt | - | - | **99.0** | - | - | - | 89.0 | 66.0 | **99.0** |
| Lage-Freitas et al. (2022) | random forest | pt | pt | - | - | 50.0 | - | - | - | 49.0 | 50.0 | 98.0 |
| Lage-Freitas et al. (2022) | support vector machine | pt | pt | - | - | 67.0 | - | - | - | 85.0 | 62.0 | 98.0 |
| lextreme | distilbert-base-multilingual-cased | pt | pt | 57.6 | 61.4 | 58.8 | 96.4 | 96.4 | 96.4 | - | - | 96.4 |
| lextreme | microsoft/mdeberta-v3-base | pt | pt | 51.9 | 57.3 | 53.2 | 96.4 | 96.4 | 96.4 | - | - | 96.4 |
| lextreme | microsoft/Multilingual-MiniLM-L12-H384 | pt | pt | 55.9 | 69.8 | 56.7 | 88.7 | 88.7 | 88.7 | - | - | 88.7 |
| lextreme | neuralmind/bert-base-portuguese-cased | pt | pt | **70.3** | **73.9** | 70.6 | 96.9 | 96.9 | 96.9 | - | - | 96.9 |
| lextreme | xlm-roberta-base | pt | pt | 61.5 | 62.1 | 61.6 | **97.7** | **97.7** | **97.7** | - | - | 97.7 |
| lextreme | xlm-roberta-large | pt | pt | 64.5 | 61.8 | 62.3 | 97.1 | 97.1 | 97.1 | - | - | 97.1 |

Table 16: BCD-U. The best scores are in bold.

| Source | Method | TrainLang | TestLang | macro-precision | macro-recall | macro-f1 | micro-precision | micro-recall | micro-f1 | precision | recall | accuracy |
|---|---|---|---|---|---|---|---|---|---|---|---|---|
| Tziafas et al. (2021) | XLM-RoBERTa | all | all | - | - | 59.2 | - | - | - | 62.6 | 60.0 | 54.6 |
| Tziafas et al. (2021) | XLM-RoBERTa pretrained on C19 | all | all | - | - | 59.8 | - | - | - | 55.9 | 62.8 | 57.7 |
| Tziafas et al. (2021) | XLM-RoBERTa pretrained on C19 | all | fr-be | - | - | 72.0 | - | - | - | **84.9** | 64.5 | - |
| Tziafas et al. (2021) | XLM-RoBERTa pretrained on C19 | all | pl | - | - | 58.3 | - | - | - | 53.3 | 66.7 | - |
| Tziafas et al. (2021) | XLM-RoBERTa pretrained on C19 | all | fr | - | - | **81.8** | - | - | - | 82.9 | **84.7** | - |
| Tziafas et al. (2021) | XLM-RoBERTa pretrained on C19 | all | it | - | - | 58.0 | - | - | - | 64.6 | 56.7 | - |
| Tziafas et al. (2021) | XLM-RoBERTa pretrained on C19 | all | nl | - | - | 55.0 | - | - | - | 62.5 | 50.0 | - |
| Tziafas et al. (2021) | XLM-RoBERTa pretrained on C19 | all | nb | - | - | 41.4 | - | - | - | 40.5 | 47.7 | - |
| Tziafas et al. (2021) | XLM-RoBERTa pretrained on C19 | all | en | - | - | 69.0 | - | - | - | 69.5 | 70.4 | - |
| Tziafas et al. (2021) | gated recurrent unit | all | all | - | - | 46.6 | - | - | - | 42.1 | 51.1 | 40.0 |
| Tziafas et al. (2021) | multi-layered perceptron | all | all | - | - | 25.7 | - | - | - | 18.5 | 50.4 | 24.7 |
| Tziafas et al. (2021) | support vector machine | all | all | - | - | 37.2 | - | - | - | 29.5 | 50.8 | 39.5 |
| Tziafas et al. (2021) | zero-shot classification XLM-RoBERTa pretrained on C19 | all without fr-be | fr-be | - | - | 43.7 | - | - | - | 55.9 | 36.6 | - |
| Tziafas et al. (2021) | zero-shot classification XLM-RoBERTa pretrained on C19 | all without pl | pl | - | - | 58.3 | - | - | - | 53.3 | 66.7 | - |
| Tziafas et al. (2021) | zero-shot classification XLM-RoBERTa pretrained on C19 | all without fr | fr | - | - | 31.8 | - | - | - | 27.0 | 39.3 | - |
| Tziafas et al. (2021) | zero-shot classification XLM-RoBERTa pretrained on C19 | all without it | it | - | - | 33.5 | - | - | - | 43.1 | 36.9 | - |
| Tziafas et al. (2021) | zero-shot classification XLM-RoBERTa pretrained on C19 | all without nl | nl | - | - | 20.6 | - | - | - | 37.5 | 23.6 | - |
| Tziafas et al. (2021) | zero-shot classification XLM-RoBERTa pretrained on C19 | all without nb | nb | - | - | 15.5 | - | - | - | 13.5 | 18.9 | - |
| Tziafas et al. (2021) | zero-shot classification XLM-RoBERTa pretrained on C19 | all without en | en | - | - | 38.4 | - | - | - | 42.3 | 37.0 | - |
| lextreme | bert-base-cased | en | en | 44.8 | 16.5 | 22.1 | 89.2 | 34.4 | 49.6 | - | - | 55.1 |
| lextreme | bert-base-uncased | en | en | 36.6 | 18.0 | 22.3 | 79.6 | 37.5 | 50.9 | - | - | 54.7 |
| lextreme | camembert-base | fr | fr | 31.5 | 9.5 | 13.7 | 77.7 | 22.0 | 34.3 | - | - | 62.5 |
| lextreme | dbmdz/bert-base-french-europeana-cased | fr | fr | 38.2 | 34.4 | 36.0 | 76.1 | **64.7** | 69.9 | - | - | 75.7 |
| lextreme | dbmdz/bert-base-italian-cased | it | it | 37.7 | 25.5 | 29.5 | 78.8 | 55.6 | 65.1 | - | - | 65.9 |
| lextreme | distilbert-base-multilingual-cased | all | all | 50.6 | 33.9 | 39.5 | 75.6 | 54.4 | 63.2 | - | - | 66.3 |
| lextreme | distilbert-base-uncased | en | en | 31.2 | 12.3 | 16.6 | 87.4 | 28.6 | 43.1 | - | - | 52.7 |
| lextreme | dkleczek/bert-base-polish-uncased-v1 | pl | pl | 20.8 | 7.8 | 11.3 | **100.0** | 29.6 | 45.5 | - | - | 36.7 |
| lextreme | dlicari/Italian-Legal-BERT | it | it | 36.3 | 18.0 | 22.3 | 81.1 | 41.5 | 54.9 | - | - | 61.3 |
| lextreme | GroNLP/bert-base-dutch-cased | nl | nl | 8.3 | 3.1 | 4.4 | 66.7 | 12.5 | 20.7 | - | - | 61.1 |
| lextreme | microsoft/mdeberta-v3-base | all | all | 50.8 | 37.9 | 40.7 | 75.6 | **64.7** | 69.8 | - | - | 69.1 |
| lextreme | microsoft/Multilingual-MiniLM-L12-H384 | all | all | 18.4 | 3.7 | 5.8 | 55.0 | 8.2 | 14.3 | - | - | 49.4 |
| lextreme | Musixmatch/umberto-commoncrawl-cased-v1 | it | it | 21.4 | 13.3 | 16.3 | 88.5 | 37.2 | 52.4 | - | - | 58.8 |
| lextreme | nlpaueb/legal-bert-base-uncased | en | en | 50.6 | 22.3 | 29.4 | 87.1 | 42.7 | 57.2 | - | - | 58.8 |
| lextreme | pdelobelle/robbert-v2-dutch-base | nl | nl | 0.0 | 0.0 | 0.0 | 0.0 | 0.0 | 0.0 | - | - | 55.6 |
| lextreme | roberta-base | en | en | 50.0 | 31.0 | 37.0 | 81.3 | 51.6 | 63.1 | - | - | 61.3 |
| lextreme | roberta-large | en | en | 40.5 | 32.1 | 35.1 | 73.3 | 53.6 | 61.9 | - | - | 56.8 |
| lextreme | SZTAKI-HLT/hubert-base-cc | hu | hu | 0.0 | 0.0 | 0.0 | 0.0 | 0.0 | 0.0 | - | - | 90.0 |
| lextreme | xlm-roberta-base | all | all | 57.0 | 41.3 | 45.9 | 74.5 | 60.6 | 66.8 | - | - | 67.8 |
| lextreme | xlm-roberta-large | all | all | **67.2** | **47.6** | 54.2 | 80.0 | 63.0 | **70.4** | - | - | 70.7 |
| lextreme | zlucia/custom-legalbert | en | en | 50.2 | 26.3 | 32.7 | 79.5 | 47.4 | 59.2 | - | - | 60.1 |

Table 17: C19. The best scores are in bold.

| Source | Method | TrainLang | TestLang | macro-precision | macro-recall | macro-f1 | micro-precision | micro-recall | micro-f1 | precision | recall | f1 | accuracy |
|---|---|---|---|---|---|---|---|---|---|---|---|---|---|
| Urchs. et al. (2021) | tf-idf/decision stump | de | de | - | - | - | - | - | - | 13.0 | 25.0 | 17.0 | 53.0 |
| Urchs. et al. (2021) | tf-idf/logistic regression | de | de | - | - | - | - | - | - | 79.0 | 63.0 | 68.0 | 77.0 |
| Urchs. et al. (2021) | tf-idf/support vector machine | de | de | - | - | - | - | - | - | 74.0 | **67.0** | **70.0** | 77.0 |
| Urchs. et al. (2021) | Unigram/decision stump | de | de | - | - | - | - | - | - | 13.0 | 25.0 | 17.0 | 53.0 |
| Urchs. et al. (2021) | Unigram/logistic regression | de | de | - | - | - | - | - | - | 74.0 | **67.0** | **70.0** | 77.0 |
| Urchs. et al. (2021) | Unigram/support vector machine | de | de | - | - | - | - | - | - | 67.0 | 66.0 | 66.0 | 74.0 |
| lextreme | dbmdz/bert-base-german-cased | de | de | 69.4 | 79.4 | 72.6 | 80.2 | 80.2 | 80.2 | - | - | - | 80.2 |
| lextreme | deepset/gbert-base | de | de | **72.6** | 80.0 | **75.1** | **82.8** | **82.8** | **82.8** | - | - | - | **82.8** |
| lextreme | distilbert-base-multilingual-cased | de | de | 66.9 | 75.9 | 69.5 | 77.9 | 77.9 | 77.9 | - | - | - | 77.9 |
| lextreme | microsoft/mdeberta-v3-base | de | de | 68.9 | **80.1** | 71.2 | 79.0 | 79.0 | 79.0 | - | - | - | 79.0 |
| lextreme | microsoft/Multilingual-MiniLM-L12-H384 | de | de | 70.6 | 78.2 | 73.3 | 80.8 | 80.8 | 80.8 | - | - | - | 80.8 |
| lextreme | xlm-roberta-base | de | de | 69.3 | 78.3 | 72.0 | 79.6 | 79.6 | 79.6 | - | - | - | 79.6 |
| lextreme | xlm-roberta-large | de | de | 70.9 | 78.9 | 73.1 | 81.0 | 81.0 | 81.0 | - | - | - | 81.0 |

Table 18: GAM. The best scores are in bold.

| Source | Method | TrainLang | TestLang | macro-precision | macro-recall | macro-f1 | micro-precision | micro-recall | micro-f1 | accuracy |
|---|---|---|---|---|---|---|---|---|---|---|
| Papaloukas et al. (2021) | BIGRU-ATT | el | el | - | - | - | 81.0 | 81.0 | 81.0 | - |
| Papaloukas et al. (2021) | BIGRU-LWAN | el | el | - | - | - | 77.0 | 77.0 | 77.0 | - |
| Papaloukas et al. (2021) | BIGRU-MAX | el | el | - | - | - | 78.0 | 78.0 | 78.0 | - |
| Papaloukas et al. (2021) | GREEK-BERT | el | el | - | - | - | 82.0 | 82.0 | 82.0 | - |
| Papaloukas et al. (2021) | GREEK-LEGAL-BERT | el | el | - | - | - | 84.0 | 84.0 | 84.0 | - |
| Papaloukas et al. (2021) | MBERT | el | el | - | - | - | 80.0 | 80.0 | 80.0 | - |
| Papaloukas et al. (2021) | Support Vector Machines + Bag-of-Words (SVM-BOW) | el | el | - | - | - | 78.0 | 78.0 | 78.0 | - |
| Papaloukas et al. (2021) | XGBOOST-BOW | el | el | - | - | - | 68.0 | 68.0 | 68.0 | - |
| Papaloukas et al. (2021) | XLM-ROBERTA | el | el | - | - | - | 81.0 | 81.0 | 81.0 | - |
| lextreme | distilbert-base-multilingual-cased | el | el | 74.7 | 69.1 | 70.0 | 80.5 | 80.5 | 80.5 | 80.5 |
| lextreme | microsoft/mdeberta-v3-base | el | el | 61.7 | 59.7 | 58.6 | 77.2 | 77.2 | 77.2 | 77.2 |
| lextreme | microsoft/Multilingual-MiniLM-L12-H384 | el | el | 42.6 | 41.2 | 39.4 | 66.1 | 66.1 | 66.1 | 66.1 |
| lextreme | nlpaueb/bert-base-greek-uncased-v1 | el | el | **78.9** | **76.5** | **76.5** | **85.3** | **85.3** | **85.3** | **85.3** |
| lextreme | xlm-roberta-base | el | el | 71.7 | 69.9 | 69.3 | 82.3 | 82.3 | 82.3 | 82.3 |
| lextreme | xlm-roberta-large | el | el | 77.2 | 74.9 | 74.7 | 84.5 | 84.5 | 84.5 | 84.5 |

Table 19: GLC-C. The best scores are in bold.

| Source | Method | TrainLang | TestLang | macro-precision | macro-recall | macro-f1 | micro-precision | micro-recall | micro-f1 | accuracy |
|---|---|---|---|---|---|---|---|---|---|---|
| Papaloukas et al. (2021) | BIGRU-ATT | el | el | - | - | - | 75.0 | 75.0 | 75.0 | - |
| Papaloukas et al. (2021) | BIGRU-LWAN | el | el | - | - | - | 65.0 | 65.0 | 65.0 | - |
| Papaloukas et al. (2021) | BIGRU-MAX | el | el | - | - | - | 63.0 | 63.0 | 63.0 | - |
| Papaloukas et al. (2021) | GREEK-BERT | el | el | - | - | - | 79.0 | 79.0 | 79.0 | - |
| Papaloukas et al. (2021) | GREEK-LEGAL-BERT | el | el | - | - | - | **81.0** | **81.0** | **81.0** | - |
| Papaloukas et al. (2021) | MBERT | el | el | - | - | - | 77.0 | 77.0 | 77.0 | - |
| Papaloukas et al. (2021) | Support Vector Machines + Bag-of-Words (SVM-BOW) | el | el | - | - | - | 38.0 | 38.0 | 38.0 | - |
| Papaloukas et al. (2021) | XGBOOST-BOW | el | el | - | - | - | 55.0 | 55.0 | 55.0 | - |
| Papaloukas et al. (2021) | XLM-ROBERTA | el | el | - | - | - | 78.0 | 78.0 | 78.0 | - |
| lextreme | distilbert-base-multilingual-cased | el | el | 34.5 | 36.8 | 33.2 | 64.4 | 64.4 | 64.4 | 64.4 |
| lextreme | microsoft/mdeberta-v3-base | el | el | 13.3 | 15.7 | 12.4 | 40.7 | 40.7 | 40.7 | 40.7 |
| lextreme | microsoft/Multilingual-MiniLM-L12-H384 | el | el | 5.5 | 7.3 | 5.2 | 28.2 | 28.2 | 28.2 | 28.2 |
| lextreme | nlpaueb/bert-base-greek-uncased-v1 | el | el | **64.0** | **65.5** | **62.8** | 80.3 | 80.3 | 80.3 | **80.3** |
| lextreme | xlm-roberta-base | el | el | 15.8 | 18.6 | 15.4 | 42.2 | 42.2 | 42.2 | 42.2 |
| lextreme | xlm-roberta-large | el | el | 39.6 | 41.3 | 39.1 | 53.6 | 53.6 | 53.6 | 53.6 |

Table 20: GLC-S. The best scores are in bold.

| Source | Method | TrainLang | TestLang | macro-precision | macro-recall | macro-f1 | micro-precision | micro-recall | micro-f1 | accuracy |
|---|---|---|---|---|---|---|---|---|---|---|
| Papaloukas et al. (2021) | BIGRU-ATT | el | el | - | - | - | 86.0 | 86.0 | 86.0 | - |
| Papaloukas et al. (2021) | BIGRU-LWAN | el | el | - | - | - | 84.0 | 84.0 | 84.0 | - |
| Papaloukas et al. (2021) | BIGRU-MAX | el | el | - | - | - | 84.0 | 84.0 | 84.0 | - |
| Papaloukas et al. (2021) | GREEK-BERT | el | el | - | - | - | 88.0 | 88.0 | 88.0 | - |
| Papaloukas et al. (2021) | GREEK-LEGAL-BERT | el | el | - | - | - | 89.0 | 89.0 | 89.0 | - |
| Papaloukas et al. (2021) | MBERT | el | el | - | - | - | 86.0 | 86.0 | 86.0 | - |
| Papaloukas et al. (2021) | Support Vector Machines + Bag-of-Words (SVM-BOW) | el | el | - | - | - | 85.0 | 85.0 | 85.0 | - |
| Papaloukas et al. (2021) | XGBOOST-BOW | el | el | - | - | - | 77.0 | 77.0 | 77.0 | - |
| Papaloukas et al. (2021) | XLM-ROBERTA | el | el | - | - | - | 85.0 | 85.0 | 85.0 | - |
| lextreme | distilbert-base-multilingual-cased | el | el | 85.8 | 84.9 | 85.2 | 87.3 | 87.3 | 87.3 | 87.3 |
| lextreme | microsoft/mdeberta-v3-base | el | el | 85.8 | 85.5 | 85.6 | 87.8 | 87.8 | 87.8 | 87.8 |
| lextreme | microsoft/Multilingual-MiniLM-L12-H384 | el | el | 82.3 | 81.6 | 81.7 | 84.8 | 84.8 | 84.8 | 84.8 |
| lextreme | nlpaueb/bert-base-greek-uncased-v1 | el | el | **88.5** | **88.0** | **88.1** | **89.8** | **89.8** | **89.8** | **89.8** |
| lextreme | xlm-roberta-base | el | el | 86.3 | 85.6 | 85.9 | 88.1 | 88.1 | 88.1 | 88.1 |
| lextreme | xlm-roberta-large | el | el | 58.4 | 59.0 | 58.3 | 62.0 | 62.0 | 62.0 | 62.0 |

Table 21: GLC-V. The best scores are in bold.

| Source | Method | TrainLang | TestLang | macro-precision | macro-recall | macro-f1 | micro-precision | micro-recall | micro-f1 | accuracy |
|---|---|---|---|---|---|---|---|---|---|---|
| Angelidis et al. (2018) | BILSTM-BILSTM-LR | el | el | **91.0** | **85.0** | **88.0** | - | - | - | - |
| Angelidis et al. (2018) | BILSTM-CRF | el | el | 87.0 | 80.0 | 83.0 | - | - | - | - |
| Angelidis et al. (2018) | BILSTM-LR | el | el | 89.0 | 79.0 | 84.0 | - | - | - | - |
| lextreme | distilbert-base-multilingual-cased | el | el | 67.5 | 76.0 | 71.0 | 97.3 | 96.8 | 97.0 | 96.6 |
| lextreme | microsoft/mdeberta-v3-base | el | el | 73.5 | 74.5 | 73.3 | 97.4 | 97.1 | 97.3 | 96.9 |
| lextreme | microsoft/Multilingual-MiniLM-L12-H384 | el | el | 75.3 | 73.2 | 74.0 | 97.5 | 97.2 | 97.3 | 96.9 |
| lextreme | nlpaueb/bert-base-greek-uncased-v1 | el | el | 74.8 | 71.4 | 72.6 | 97.3 | 97.0 | 97.1 | 96.8 |
| lextreme | xlm-roberta-base | el | el | 75.7 | 73.8 | 74.6 | 97.5 | **97.3** | **97.4** | 97.0 |
| lextreme | xlm-roberta-large | el | el | 73.1 | 76.3 | 74.1 | **97.6** | **97.3** | **97.4** | **97.1** |

Table 22: GLN. The best scores are in bold.

| Source | Method | TrainLang | TestLang | macro-precision | macro-recall | macro-f1 | micro-precision | micro-recall | micro-f1 | accuracy |
|---|---|---|---|---|---|---|---|---|---|---|
| Pais et al. (2021) | CoRoLa word embeddings + MARCELL word embeddings+BiLSTM-CRF | ro | ro | - | - | 84.7 | - | - | - | - |
| Pais et al. (2021) | CoRoLa word embeddings + MARCELL word embeddings+BiLSTM-CRF + gazetteers | ro | ro | - | - | 84.8 | - | - | - | - |
| Pais et al. (2021) | CoRoLa word embeddings + MARCELL word embeddings+BiLSTM-CRF + gazetteers + affixes | ro | ro | - | - | 83.4 | - | - | - | - |
| Pais et al. (2021) | CoRoLa word embeddings+BiLSTM-CRF | ro | ro | - | - | 83.9 | - | - | - | - |
| Pais et al. (2021) | CoRoLa word embeddings+BiLSTM-CRF + gazetteers | ro | ro | - | - | 85.0 | - | - | - | - |
| Pais et al. (2021) | CoRoLa word embeddings+BiLSTM-CRF + gazetteers + affixes | ro | ro | - | - | 83.9 | - | - | - | - |
| Pais et al. (2021) | Intersection | ro | ro | - | - | 86.1 | - | - | - | - |
| Pais et al. (2021) | Longest span | ro | ro | - | - | 87.3 | - | - | - | - |
| Pais et al. (2021) | MARCELL word embeddings+BiLSTM-CRF | ro | ro | - | - | 83.5 | - | - | - | - |
| Pais et al. (2021) | MARCELL word embeddings+BiLSTM-CRF + gazetteers | ro | ro | - | - | 85.3 | - | - | - | - |
| Pais et al. (2021) | MARCELL word embeddings+BiLSTM-CRF + gazetteers + affixes | ro | ro | - | - | 83.4 | - | - | - | - |
| Pais et al. (2021) | Reunion | ro | ro | - | - | **90.4** | - | - | - | - |
| Pais et al. (2021) | Voting algorithm | ro | ro | - | - | 89.4 | - | - | - | - |
| lextreme | distilbert-base-multilingual-cased | ro | ro | 85.5 | 85.1 | 85.3 | 97.3 | 96.6 | 96.9 | 96.4 |
| lextreme | dumitrescustefan/bert-base-romanian-uncased-v1 | ro | ro | 85.2 | 82.4 | 83.6 | 96.8 | 96.8 | 96.8 | 96.2 |
| lextreme | microsoft/mdeberta-v3-base | ro | ro | **85.9** | 84.6 | 85.1 | 97.4 | 96.8 | 97.1 | 96.7 |
| lextreme | microsoft/Multilingual-MiniLM-L12-H384 | ro | ro | 85.2 | 83.9 | 84.5 | 97.4 | 97.1 | 97.2 | 96.8 |
| lextreme | readerbench/jurBERT-base | ro | ro | 83.1 | 86.6 | 84.7 | 97.0 | **97.2** | 97.1 | 96.6 |
| lextreme | xlm-roberta-base | ro | ro | 84.9 | **87.0** | 85.8 | 97.4 | **97.2** | **97.3** | **96.9** |
| lextreme | xlm-roberta-large | ro | ro | 85.0 | 85.3 | 85.0 | **97.5** | 97.2 | **97.3** | **96.9** |

Table 23: LNR. The best scores are in bold.

| Source | Method | TrainLang | TestLang | macro-precision | macro-recall | macro-f1 | micro-precision | micro-recall | micro-f1 | precision | recall | f1 | accuracy |
|---|---|---|---|---|---|---|---|---|---|---|---|---|---|
| Luz de Araujo et al. (2018) | LSTM-CRF (long short-term memory + conditional random field) | pt | pt | - | - | - | - | - | - | **93.2** | 91.9 | 92.5 | - |
| lextreme | distilbert-base-multilingual-cased | pt | pt | 89.9 | 89.6 | 89.6 | 98.5 | 98.6 | 98.6 | - | - | - | 98.0 |
| lextreme | microsoft/mdeberta-v3-base | pt | pt | 94.7 | 94.9 | 94.8 | 99.0 | **99.4** | **99.2** | - | - | - | 98.8 |
| lextreme | microsoft/Multilingual-MiniLM-L12-H384 | pt | pt | 93.9 | 93.4 | 93.6 | 98.9 | 99.2 | 99.1 | - | - | - | 98.7 |
| lextreme | neuralmind/bert-base-portuguese-cased | pt | pt | 94.7 | 93.5 | 94.1 | 99.0 | 99.2 | 99.1 | - | - | - | 98.8 |
| lextreme | xlm-roberta-base | pt | pt | 94.3 | 94.0 | 94.1 | 99.0 | 99.2 | 99.1 | - | - | - | 98.7 |
| lextreme | xlm-roberta-large | pt | pt | **95.5** | **95.2** | **95.3** | **99.1** | **99.4** | **99.2** | - | - | - | **98.9** |

Table 24: LNB. The best scores are in bold.

| Source | Method | TrainLang | TestLang | macro-precision | macro-recall | macro-f1 | micro-precision | micro-recall | micro-f1 | accuracy | mean r-precision |
|---|---|---|---|---|---|---|---|---|---|---|---|
| Chalkidis et al. (2021a) | xlm-roberta-base | all | all | - | - | - | - | - | - | - | 65.7 |
| Chalkidis et al. (2021a) | xlm-roberta-base | all | en | - | - | - | - | - | - | - | 66.4 |
| Chalkidis et al. (2021a) | xlm-roberta-base | all | da | - | - | - | - | - | - | - | 66.2 |
| Chalkidis et al. (2021a) | xlm-roberta-base | all | de | - | - | - | - | - | - | - | 66.2 |
| Chalkidis et al. (2021a) | xlm-roberta-base | all | nl | - | - | - | - | - | - | - | 66.1 |
| Chalkidis et al. (2021a) | xlm-roberta-base | all | sv | - | - | - | - | - | - | - | 66.1 |
| Chalkidis et al. (2021a) | xlm-roberta-base | all | ro | - | - | - | - | - | - | - | 66.3 |
| Chalkidis et al. (2021a) | xlm-roberta-base | all | es | - | - | - | - | - | - | - | 66.3 |
| Chalkidis et al. (2021a) | xlm-roberta-base | all | fr | - | - | - | - | - | - | - | 66.2 |
| Chalkidis et al. (2021a) | xlm-roberta-base | all | it | - | - | - | - | - | - | - | 66.3 |
| Chalkidis et al. (2021a) | xlm-roberta-base | all | pt | - | - | - | - | - | - | - | 65.9 |
| Chalkidis et al. (2021a) | xlm-roberta-base | all | bg | - | - | - | - | - | - | - | 65.7 |
| Chalkidis et al. (2021a) | xlm-roberta-base | all | cs | - | - | - | - | - | - | - | 65.7 |
| Chalkidis et al. (2021a) | xlm-roberta-base | all | hr | - | - | - | - | - | - | - | 65.8 |
| Chalkidis et al. (2021a) | xlm-roberta-base | all | pl | - | - | - | - | - | - | - | 65.6 |
| Chalkidis et al. (2021a) | xlm-roberta-base | all | sk | - | - | - | - | - | - | - | 65.7 |
| Chalkidis et al. (2021a) | xlm-roberta-base | all | sl | - | - | - | - | - | - | - | 65.8 |
| Chalkidis et al. (2021a) | xlm-roberta-base | all | hu | - | - | - | - | - | - | - | 65.2 |
| Chalkidis et al. (2021a) | xlm-roberta-base | all | fi | - | - | - | - | - | - | - | 65.8 |
| Chalkidis et al. (2021a) | xlm-roberta-base | all | et | - | - | - | - | - | - | - | 65.6 |
| Chalkidis et al. (2021a) | xlm-roberta-base | all | lt | - | - | - | - | - | - | - | 65.7 |
| Chalkidis et al. (2021a) | xlm-roberta-base | all | lv | - | - | - | - | - | - | - | 65.8 |
| Chalkidis et al. (2021a) | xlm-roberta-base | all | el | - | - | - | - | - | - | - | 65.1 |
| Chalkidis et al. (2021a) | xlm-roberta-base | all | mt | - | - | - | - | - | - | - | 62.3 |
| Chalkidis et al. (2021a) | xlm-roberta-base + Adapters layers | all | all | - | - | - | - | - | - | - | 66.4 |
| Chalkidis et al. (2021a) | xlm-roberta-base + Adapters layers | all | en | - | - | - | - | - | - | - | 67.3 |
| Chalkidis et al. (2021a) | xlm-roberta-base + Adapters layers | all | da | - | - | - | - | - | - | - | 67.1 |
| Chalkidis et al. (2021a) | xlm-roberta-base + Adapters layers | all | de | - | - | - | - | - | - | - | 66.3 |
| Chalkidis et al. (2021a) | xlm-roberta-base + Adapters layers | all | nl | - | - | - | - | - | - | - | 67.1 |
| Chalkidis et al. (2021a) | xlm-roberta-base + Adapters layers | all | sv | - | - | - | - | - | - | - | 67.0 |
| Chalkidis et al. (2021a) | xlm-roberta-base + Adapters layers | all | ro | - | - | - | - | - | - | - | **67.4** |
| Chalkidis et al. (2021a) | xlm-roberta-base + Adapters layers | all | es | - | - | - | - | - | - | - | 67.2 |
| Chalkidis et al. (2021a) | xlm-roberta-base + Adapters layers | all | fr | - | - | - | - | - | - | - | 67.1 |
| Chalkidis et al. (2021a) | xlm-roberta-base + Adapters layers | all | it | - | - | - | - | - | - | - | **67.4** |
| Chalkidis et al. (2021a) | xlm-roberta-base + Adapters layers | all | pt | - | - | - | - | - | - | - | 67.0 |
| Chalkidis et al. (2021a) | xlm-roberta-base + Adapters layers | all | bg | - | - | - | - | - | - | - | 66.6 |
| Chalkidis et al. (2021a) | xlm-roberta-base + Adapters layers | all | cs | - | - | - | - | - | - | - | 67.0 |
| Chalkidis et al. (2021a) | xlm-roberta-base + Adapters layers | all | hr | - | - | - | - | - | - | - | 67.0 |
| Chalkidis et al. (2021a) | xlm-roberta-base + Adapters layers | all | pl | - | - | - | - | - | - | - | 66.2 |
| Chalkidis et al. (2021a) | xlm-roberta-base + Adapters layers | all | sk | - | - | - | - | - | - | - | 66.2 |
| Chalkidis et al. (2021a) | xlm-roberta-base + Adapters layers | all | sl | - | - | - | - | - | - | - | 66.8 |
| Chalkidis et al. (2021a) | xlm-roberta-base + Adapters layers | all | hu | - | - | - | - | - | - | - | 65.5 |
| Chalkidis et al. (2021a) | xlm-roberta-base + Adapters layers | all | fi | - | - | - | - | - | - | - | 66.6 |
| Chalkidis et al. (2021a) | xlm-roberta-base + Adapters layers | all | et | - | - | - | - | - | - | - | 65.7 |
| Chalkidis et al. (2021a) | xlm-roberta-base + Adapters layers | all | lt | - | - | - | - | - | - | - | 65.8 |
| Chalkidis et al. (2021a) | xlm-roberta-base + Adapters layers | all | lv | - | - | - | - | - | - | - | 66.7 |
| Chalkidis et al. (2021a) | xlm-roberta-base + Adapters layers | all | el | - | - | - | - | - | - | - | 65.7 |
| Chalkidis et al. (2021a) | xlm-roberta-base + Adapters layers | all | mt | - | - | - | - | - | - | - | 61.6 |
| lextreme | bert-base-cased | en | en | 21.1 | 42.2 | 25.5 | 40.7 | 75.2 | 52.8 | 0.5 | - |
| lextreme | bert-base-uncased | en | en | 21.7 | 39.8 | 25.3 | 41.9 | 73.8 | 53.4 | 0.8 | - |
| lextreme | bertin-project/bertin-roberta-base-spanish | es | es | 26.0 | 12.8 | 15.4 | 79.1 | 46.0 | 58.2 | 2.7 | - |
| lextreme | camembert-base | fr | fr | 23.2 | 16.9 | 17.4 | 70.3 | 56.3 | 62.5 | 2.1 | - |
| lextreme | dbmdz/bert-base-french-europeana-cased | fr | fr | 21.6 | 33.5 | 23.7 | 44.8 | 69.6 | 54.5 | 1.4 | - |
| lextreme | dbmdz/bert-base-german-cased | de | de | 21.7 | 39.0 | 25.0 | 42.6 | 73.5 | 53.9 | 0.7 | - |
| lextreme | dbmdz/bert-base-italian-cased | it | it | 20.0 | 36.6 | 23.1 | 41.5 | 72.6 | 52.8 | 0.4 | - |
| lextreme | deepset/gbert-base | de | de | 20.2 | 36.8 | 23.6 | 41.8 | 72.4 | 53.0 | 0.6 | - |
| lextreme | distilbert-base-multilingual-cased | all | all | 15.9 | 13.1 | 12.0 | 55.9 | 51.9 | 53.8 | 1.6 | - |
| lextreme | distilbert-base-uncased | en | en | **30.6** | 29.6 | **27.6** | 61.9 | 65.7 | **63.7** | 1.9 | - |
| lextreme | dkleczek/bert-base-polish-uncased-v1 | pl | pl | 23.2 | 31.4 | 23.4 | 45.9 | 69.4 | 55.3 | 0.7 | - |
| lextreme | dlicari/Italian-Legal-BERT | it | it | 16.0 | 31.5 | 18.8 | 37.5 | 70.8 | 48.8 | 0.4 | - |
| lextreme | dumitrescustefan/bert-base-romanian-uncased-v1 | ro | ro | 23.7 | 31.0 | 23.5 | 48.2 | 69.6 | 56.9 | 0.8 | - |
| lextreme | gerulata/slovakbert | sk | sk | 16.8 | 11.1 | 11.9 | 73.8 | 48.5 | 58.5 | 1.5 | - |
| lextreme | GroNLP/bert-base-dutch-cased | nl | nl | 26.0 | 26.3 | 23.4 | 57.1 | 64.6 | 60.6 | **3.7** | - |
| lextreme | iarfmoose/roberta-base-bulgarian | bg | bg | 4.6 | 2.9 | 3.2 | 38.7 | 18.2 | 24.8 | 0.6 | - |
| lextreme | KB/bert-base-swedish-cased | sv | sv | 23.5 | 35.1 | 25.1 | 46.4 | 72.2 | 56.5 | 0.8 | - |
| lextreme | Maltehb/danish-bert-botxo | da | da | 22.2 | 34.6 | 24.3 | 45.1 | 72.0 | 55.4 | 0.9 | - |
| lextreme | microsoft/mdeberta-v3-base | all | all | 12.7 | 19.2 | 13.1 | 39.7 | 62.0 | 48.1 | 0.2 | - |
| lextreme | microsoft/Multilingual-MiniLM-L12-H384 | all | all | 8.4 | 4.9 | 5.6 | 76.2 | 30.2 | 43.3 | 0.9 | - |
| lextreme | Musixmatch/umberto-commoncrawl-cased-v1 | it | it | 25.7 | 19.6 | 20.3 | 68.9 | 57.9 | 62.9 | 2.5 | - |
| lextreme | neuralmind/bert-base-portuguese-cased | pt | pt | 21.8 | 33.2 | 23.5 | 45.5 | 70.5 | 55.3 | 1.0 | - |
| lextreme | nlpaueb/bert-base-greek-uncased-v1 | el | el | 19.8 | 39.0 | 23.5 | 40.5 | 73.1 | 52.1 | 0.5 | - |
| lextreme | nlpaueb/legal-bert-base-uncased | en | en | 19.6 | **43.4** | 24.3 | 38.6 | **77.0** | 51.4 | 0.4 | - |
| lextreme | pdelobelle/robbert-v2-dutch-base | nl | nl | 19.3 | 15.3 | 15.3 | 66.0 | 54.9 | 60.0 | 1.6 | - |
| lextreme | PlanTL-GOB-ES/roberta-base-bne | es | es | 21.8 | 10.9 | 13.1 | **80.5** | 45.2 | 57.9 | 2.2 | - |
| lextreme | PlanTL-GOB-ES/RoBERTalex | es | es | 23.3 | 21.9 | 20.1 | 58.9 | 62.5 | 60.6 | 1.3 | - |
| lextreme | readerbench/jurBERT-base | ro | ro | 21.0 | 25.7 | 20.8 | 48.4 | 65.5 | 55.7 | 0.8 | - |
| lextreme | roberta-base | en | en | 0.0 | 0.0 | 0.0 | 0.0 | 0.0 | 0.0 | 0.3 | - |
| lextreme | roberta-large | en | en | 12.8 | 17.6 | 13.3 | 37.4 | 35.9 | 31.1 | 0.9 | - |
| lextreme | SZTAKI-HLT/hubert-base-cc | hu | hu | 25.6 | 30.2 | 24.7 | 50.8 | 66.7 | 57.7 | 1.1 | - |
| lextreme | tartuNLP/EstBERT | et | et | 20.0 | 31.7 | 21.9 | 43.6 | 69.3 | 53.5 | 0.5 | - |
| lextreme | TurkuNLP/bert-base-finnish-cased-v1 | fi | fi | 22.9 | 35.4 | 25.1 | 46.6 | 70.8 | 56.2 | 0.8 | - |
| lextreme | UWB-AIR/Czert-B-base-cased | cs | cs | 23.4 | 35.7 | 25.2 | 45.4 | 70.9 | 55.3 | 0.6 | - |
| lextreme | xlm-roberta-base | all | all | 7.1 | 4.2 | 4.7 | 77.2 | 30.6 | 43.8 | 0.7 | - |
| lextreme | xlm-roberta-large | all | all | 19.5 | 14.3 | 14.2 | 66.9 | 54.5 | 60.0 | 1.2 | - |
| lextreme | zlucia/custom-legalbert | en | en | 19.6 | 36.8 | 22.6 | 40.4 | 73.7 | 52.1 | 0.4 | - |

Table 25: MEU-3. The best scores are in bold.

| Source | Method | TrainLang | TestLang | macro-precision | macro-recall | macro-f1 | micro-precision | micro-recall | micro-f1 | accuracy |
|---|---|---|---|---|---|---|---|---|---|---|
| Niklaus et al. (2021) | French BERT | fr | fr | - | - | 58.6 | - | - | 74.7 | - |
| Niklaus et al. (2021) | French Hierarchical BERT | fr | fr | - | - | **70.2** | - | - | 80.2 | - |
| Niklaus et al. (2021) | French Long BERT | fr | fr | - | - | 68.0 | - | - | 77.2 | - |
| Niklaus et al. (2021) | German Hierarchical BERT | de | de | - | - | 68.5 | - | - | 77.1 | - |
| Niklaus et al. (2021) | German Long BERT | de | de | - | - | 67.9 | - | - | 76.5 | - |
| Niklaus et al. (2021) | German-BERT | de | de | - | - | 63.7 | - | - | 74.0 | - |
| Niklaus et al. (2021) | Italian BERT | it | it | - | - | 55.2 | - | - | 76.1 | - |
| Niklaus et al. (2021) | Italian Hierarchical BERT | it | it | - | - | 57.1 | - | - | 75.8 | - |
| Niklaus et al. (2021) | Italian Long BERT | it | it | - | - | 59.8 | - | - | 77.1 | - |
| Niklaus et al. (2021) | Multilingual BERT | de | de | - | - | 58.2 | - | - | 68.4 | - |
| Niklaus et al. (2021) | Multilingual BERT | fr | fr | - | - | 55.0 | - | - | 71.3 | - |
| Niklaus et al. (2021) | Multilingual BERT | it | it | - | - | 53.0 | - | - | 77.6 | - |
| Niklaus et al. (2021) | Multilingual Hierarchical BERT | de | de | - | - | 57.1 | - | - | 76.8 | - |
| Niklaus et al. (2021) | Multilingual Hierarchical BERT | fr | fr | - | - | 67.2 | - | - | 76.3 | - |
| Niklaus et al. (2021) | Multilingual Hierarchical BERT | it | it | - | - | 55.5 | - | - | 72.4 | - |
| Niklaus et al. (2021) | Multilingual Long BERT | de | de | - | - | 66.5 | - | - | 75.9 | - |
| Niklaus et al. (2021) | Multilingual Long BERT | fr | fr | - | - | 64.3 | - | - | 73.3 | - |
| Niklaus et al. (2021) | Multilingual Long BERT | it | it | - | - | 58.4 | - | - | 76.0 | - |
| lextreme | camembert-base | fr | fr | 68.2 | **73.3** | 69.7 | 78.9 | 78.9 | 78.9 | 78.9 |
| lextreme | dbmdz/bert-base-french-europeana-cased | fr | fr | **69.7** | 72.2 | **70.2** | **80.4** | **80.4** | **80.4** | **80.4** |
| lextreme | dbmdz/bert-base-german-cased | de | de | 67.5 | 72.0 | 68.7 | 77.3 | 77.3 | 77.3 | 77.3 |
| lextreme | dbmdz/bert-base-italian-cased | it | it | 63.3 | 56.8 | 57.3 | 79.8 | 79.8 | 79.8 | 79.8 |
| lextreme | deepset/gbert-base | de | de | 68.6 | 71.6 | 69.3 | 78.7 | 78.7 | 78.7 | 78.7 |
| lextreme | distilbert-base-multilingual-cased | all | all | 65.8 | 72.3 | 66.7 | 74.3 | 74.3 | 74.3 | 74.3 |
| lextreme | dlicari/Italian-Legal-BERT | it | it | 65.1 | 60.2 | 60.6 | 80.3 | 80.3 | 80.3 | 80.3 |
| lextreme | microsoft/mdeberta-v3-base | all | all | 67.7 | 71.4 | 69.0 | 78.6 | 78.6 | 78.6 | 78.6 |
| lextreme | microsoft/Multilingual-MiniLM-L12-H384 | all | all | 66.4 | 71.9 | 67.6 | 76.1 | 76.1 | 76.1 | 76.1 |
| lextreme | Musixmatch/umberto-commoncrawl-cased-v1 | it | it | 62.1 | 56.9 | 57.4 | 78.4 | 78.4 | 78.4 | 78.4 |
| lextreme | xlm-roberta-base | all | all | 67.1 | 72.9 | 68.3 | 76.4 | 76.4 | 76.4 | 76.4 |
| lextreme | xlm-roberta-large | all | all | 67.1 | 72.5 | 68.3 | 76.6 | 76.6 | 76.6 | 76.6 |

Table 26: SJP. The best scores are in bold.

## H Histograms

In the following, we provide the histograms for the distribution of the sequence length of the input (sentence or entire document) from each dataset. The length is measured by counting the tokens using the tokenizers of the multilingual models, i.e., DistilBERT, MiniLM, mDeBERTa v3, XLM-R base, XLM-R large. We only display the distribution within the 99th percentile; the rest is grouped together at the end.

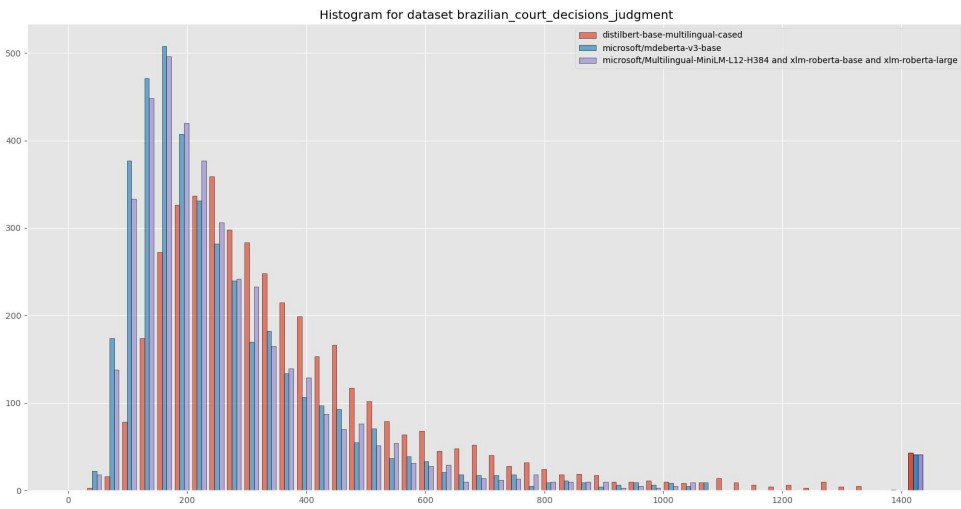

Figure 2: Histogram for dataset BCD-J

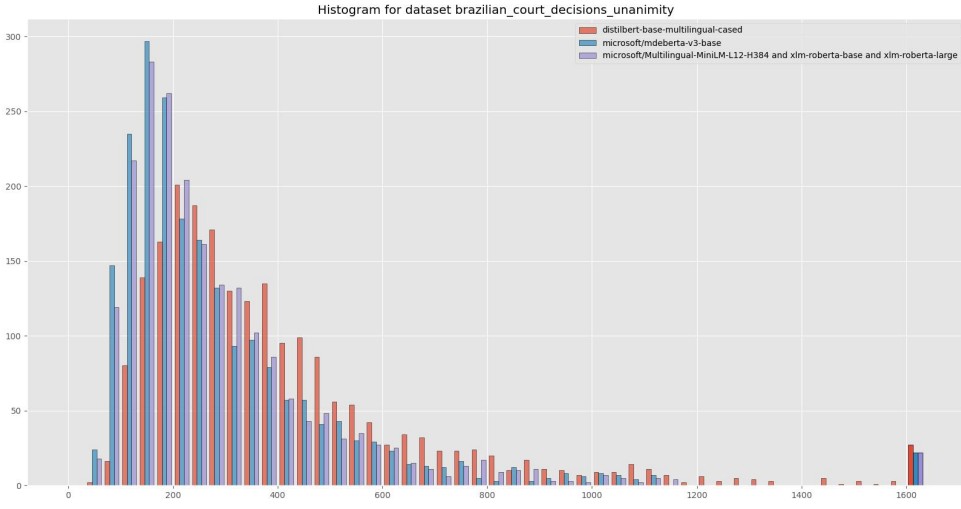

Figure 3: Histogram for dataset BCD-U

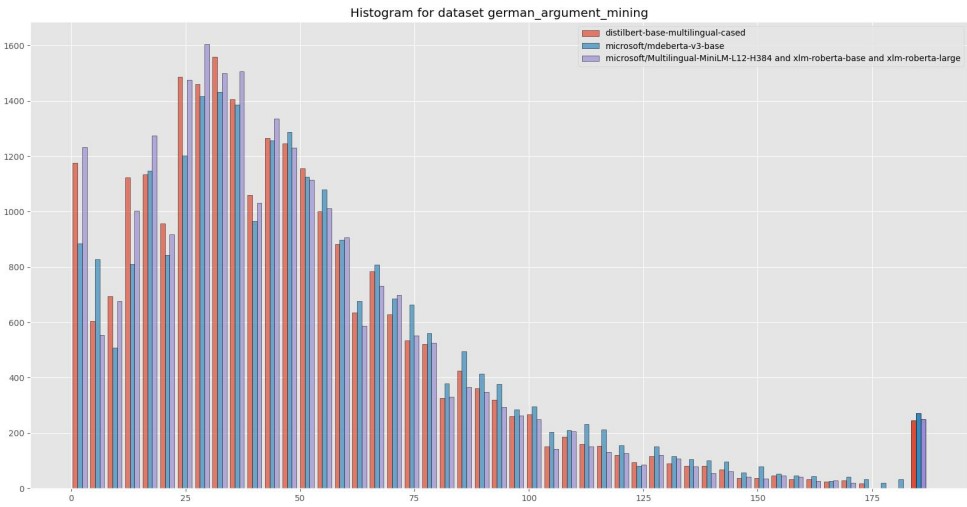

Figure 4: Histogram for dataset GAM

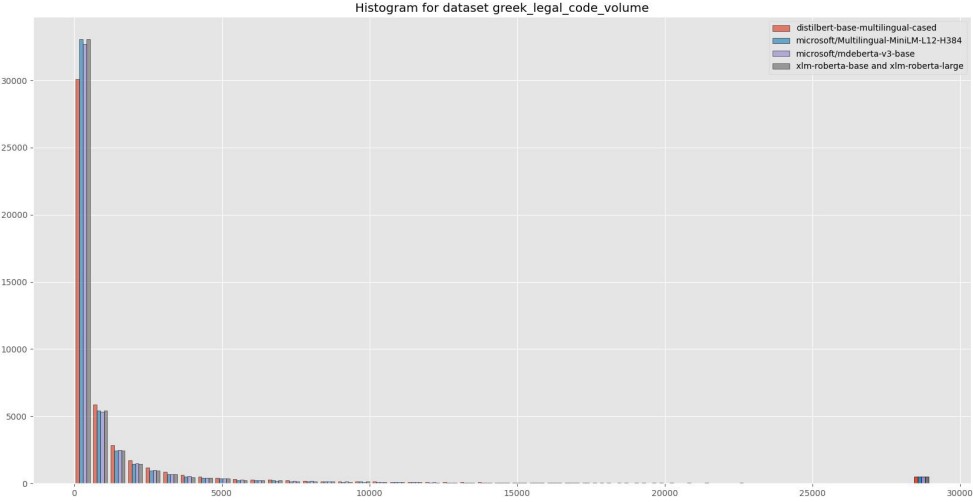

Figure 5: Histogram for dataset GLC-V

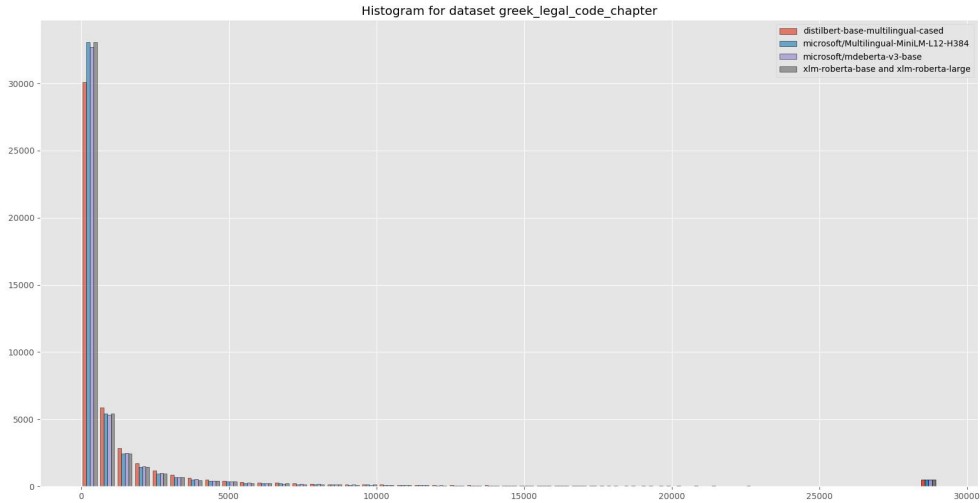

Figure 6: Histogram for dataset GLC-C

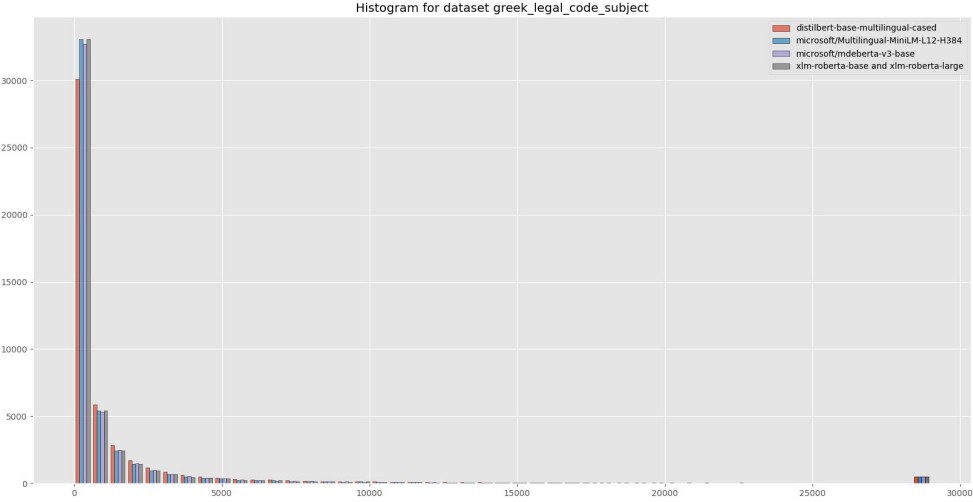

Figure 7: Histogram for dataset GLC-S

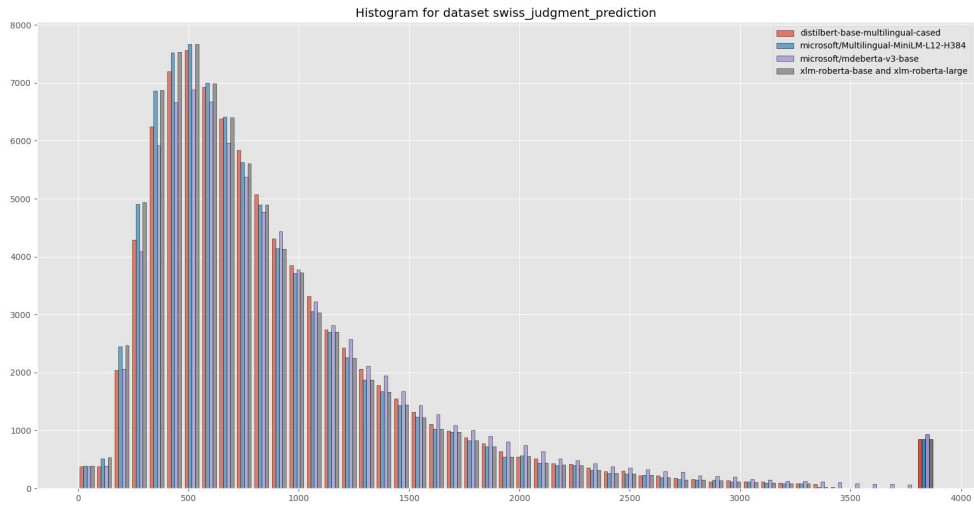

Figure 8: Histogram for dataset SJP

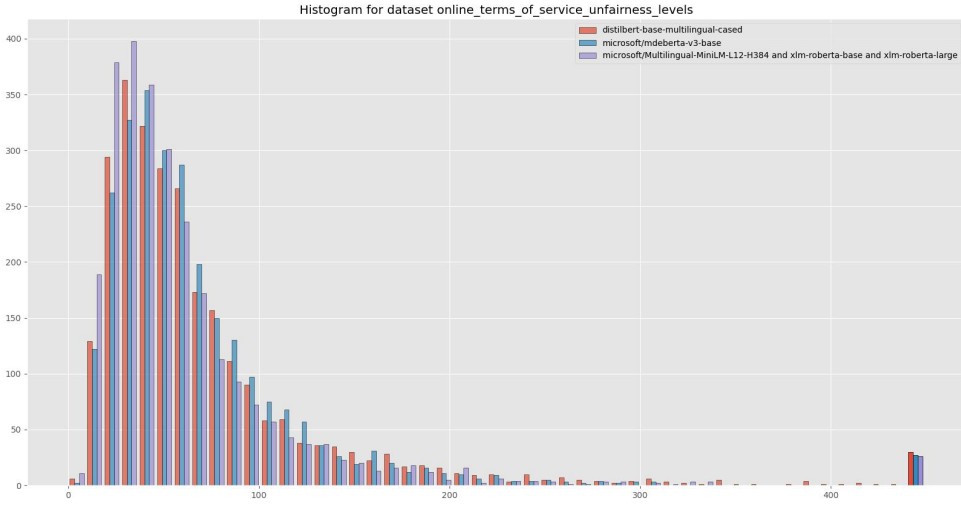

Figure 9: Histogram for dataset OTS-UL

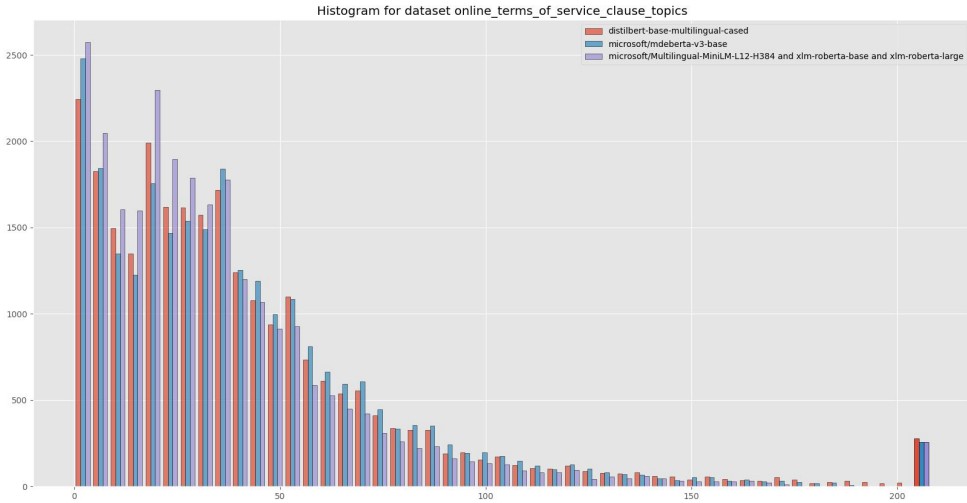

Figure 10: Histogram for dataset OTS-CT

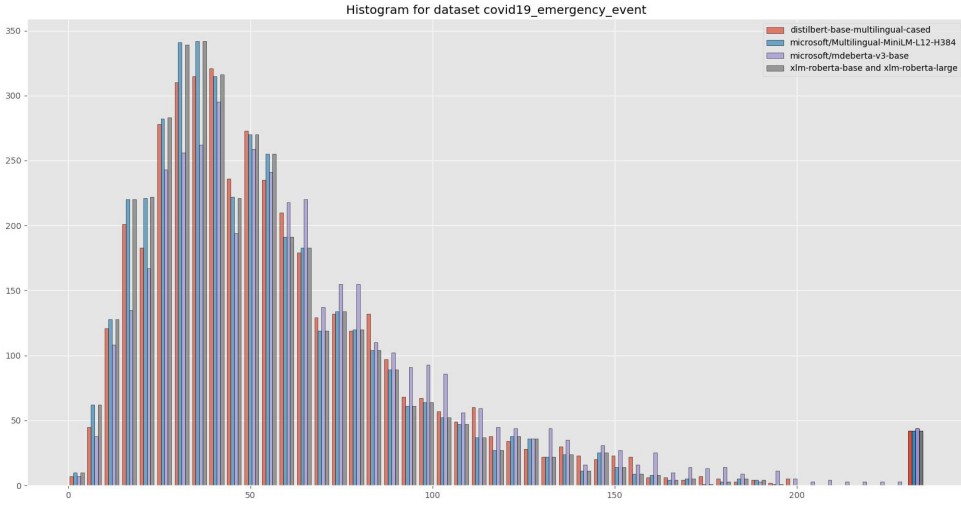

Figure 11: Histogram for dataset C19

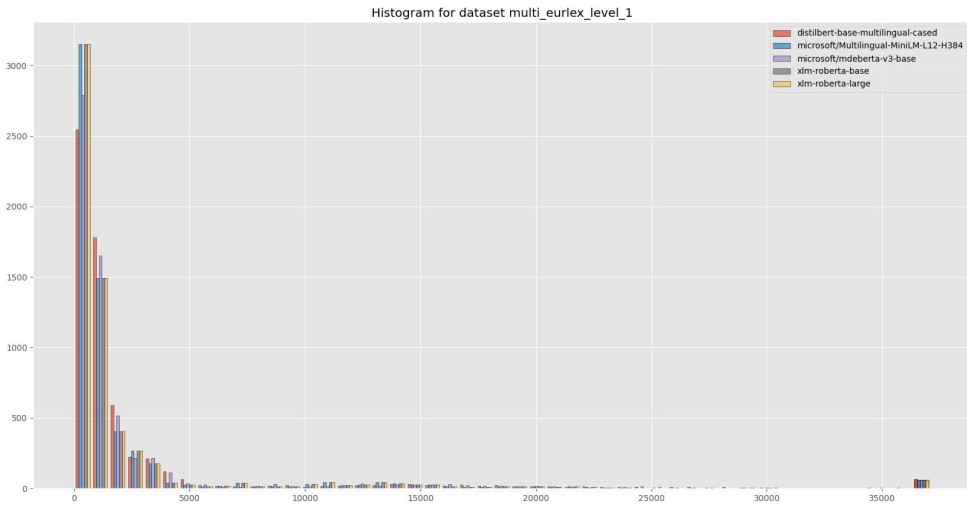

Figure 12: Histogram for dataset MEU-1

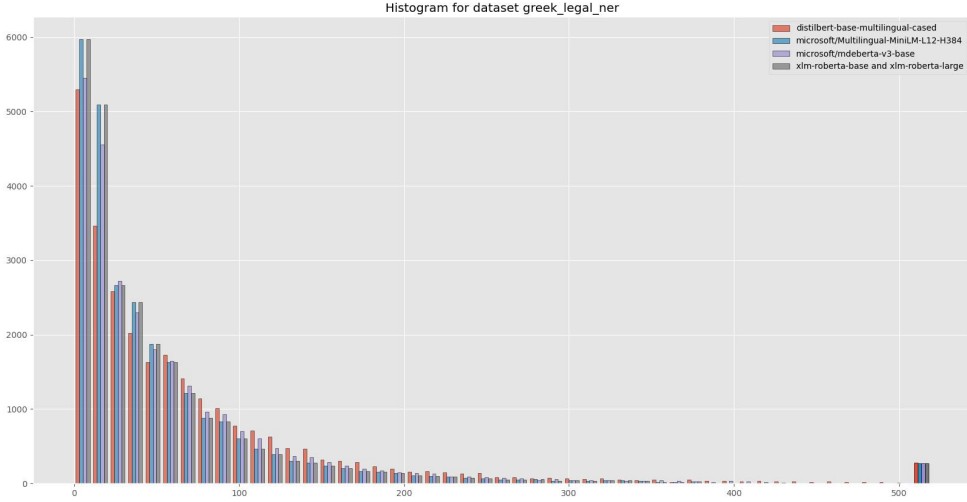

Figure 13: Histogram for dataset GLN

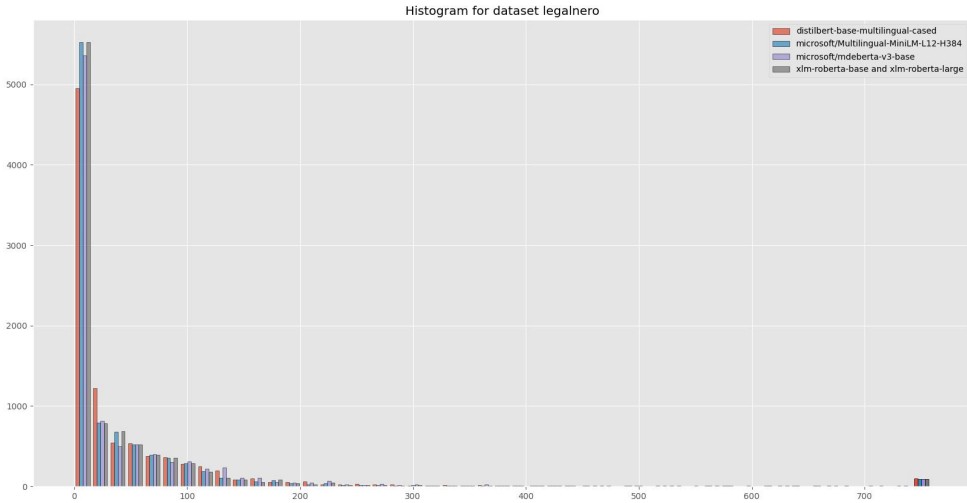

Figure 14: Histogram for dataset LNR

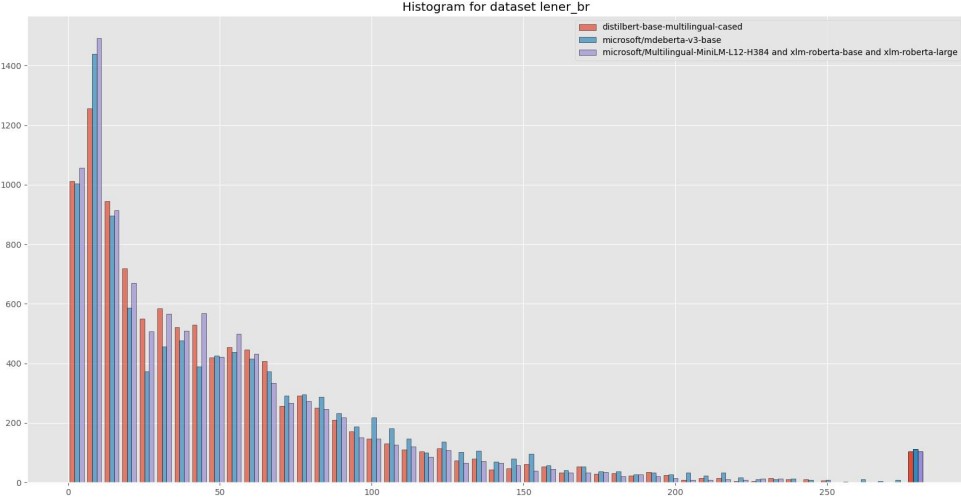

Figure 15: Histogram for dataset LNB

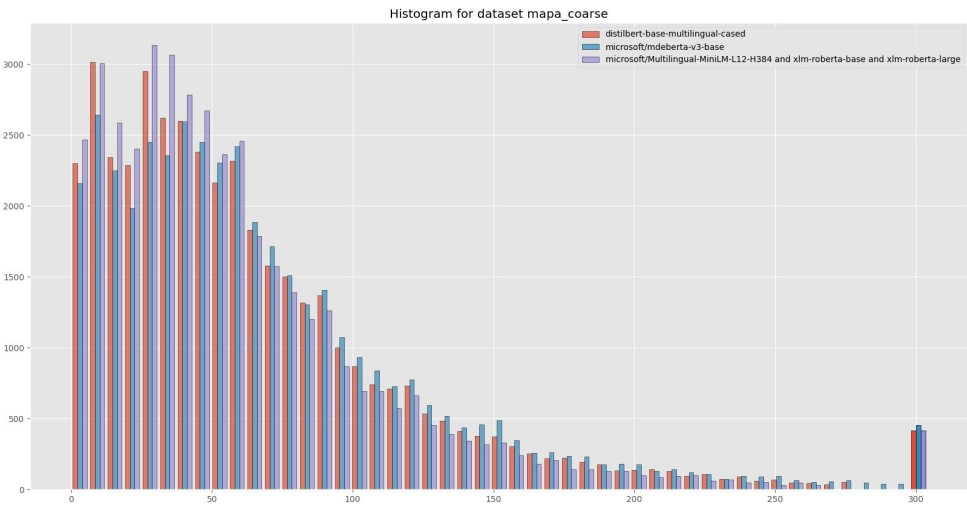

Figure 16: Histogram for dataset MAP-C

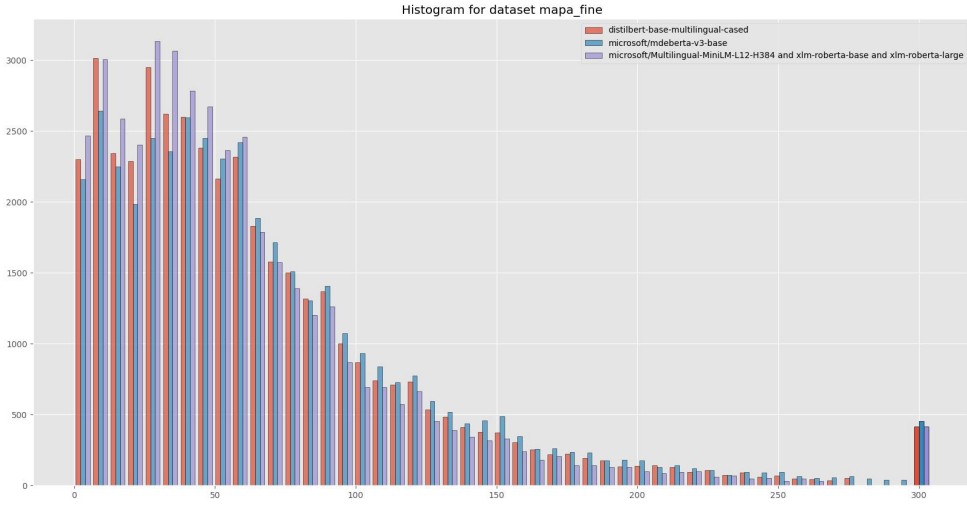

Figure 17: Histogram for dataset MAP-F