# OpenReview forum: "LEXTREME: A Multi-Lingual and Multi-Task Benchmark for the Legal Domain"
_EMNLP/2023/Conference — EMNLP 2023 Findings_

### Official Review · Reviewer_CdWt · 2023-08-02

**Soundness:** 4

**Excitement:**

4: Strong: This paper deepens the understanding of some phenomenon or lowers the barriers to an existing research direction.

**Paper Topic And Main Contributions:**

In this paper, authors introduce a new benchmark for legal NLP namely LEXTREME. It contains 11 datasets across 24 languages. To compare state-of-the-art models authors propose two different aggregate scores i.e., dataset aggregate score and language aggregate score. Authors also release LEXTREME on huggingface along with a public leaderboard and  necessary codes.

**Reasons To Accept:**

The paper is well-written and very much comprehensive. This paper has a contribution which will help the legal NLP community significantly.

**Reasons To Reject:**

N/A

**Reproducibility:**

5: Could easily reproduce the results.

**Reviewer Confidence:**

4: Quite sure. I tried to check the important points carefully. It's unlikely, though conceivable, that I missed something that should affect my ratings.

---

> ### Author Rebuttal · Authors · 2023-08-29
>
> We thank CdWt for their review. We appreciate that they are convinced that LEXTREME will help the legal NLP community significantly and that the paper is well-written and comprehensive.

---

### Official Review · Reviewer_RRMe · 2023-08-04

**Soundness:** 3

**Excitement:**

3: Ambivalent: It has merits (e.g., it reports state-of-the-art results, the idea is nice), but there are key weaknesses (e.g., it describes incremental work), and it can significantly benefit from another round of revision. However, I won't object to accepting it if my co-reviewers champion it.

**Paper Topic And Main Contributions:**

The paper presents a collection of (already) public legal text datasets, in order to build an open multilingual and multi-task benchmark, LEXTREME. The main contributions of this work are bringing together a diverse set of datasets from multiple sources into a single place, providing an integration with the popular huggingface. Finally, the authors evaluate a plethora of models on the entire collection of tasks and datasets.

**Questions For The Authors:**

A. The results (discussion) section is a bit underwhelming. While interesting observations are made there are little to no possible explanations for the results.

B. Do you have any insights on why native monolingual models are so strong compared to native legal models? Why does it seem that (pre)training on legal data leads to worse models? Is there a significant difference based on task type (i.e NER vs SLTC/MLTC)?

C. How does the jurisdiction influence the results? I think an analysis performed also on jurisdiction is almost mandatory, as different countries (and therefore languages) have widely different jurisdiction. As your work shows, at the current time, the best solution for a particular language is a native (legal or general) model. Maybe a multimodal model trained in the same jurisdiction (e.g. French law system) could better understand and reason over legal texts.

D. I find the last remark in the limitation section very important to the practical value of the benchmark. Already having access to the result of the case and the judge’s reasoning kind of beats the purpose. Furthermore the documents provided by the judge can (and most probably do) contain elements that explain and emphasise the final decision. Finally, in most cases, the documents from the judge are significantly smaller in size (number of tokens) than the documents brought forward by the involved parties. With all these simplifications, the task is still hard for current models.

E. What is your intuition, how does a “better” model look? More data? More quality data? Larger models?


**Reasons To Accept:**

- This work brings together, in a single accessible space most of the open datasets in the legal domain. This leads to a new multilingual and multi-task benchmark for the legal domain. Furthermore, this is a living benchmark, with a public leaderboard and full access to code to run experiments.

- As for the results, LEXTREME proves that, at least for the moment, a good multilingual legal model does not exist, as monolingual general and legal models perform better. There seems to be a good margin for improvement on most tasks, as they are pretty challenging.


**Reasons To Reject:**

- There is nothing actually novel from a research perspective: all datasets are already public, there is no new model or architecture presented. While the effort is definitely present and the contributions are nice to have and definitely useful for the legal domain, I don’t see any novelty worthy of this conference.

- As legal documents are usually long, as the authors also mention, I would have liked to see a more detailed discussion on how recent works on long context can improve the results - e.g. SLED [1] and others that can be applied to any transformer encoder.

[1] - Ivgi, M., Shaham, U., & Berant, J. (2023). Efficient long-text understanding with short-text models. Transactions of the Association for Computational Linguistics, 11, 284-299.

**Reproducibility:**

5: Could easily reproduce the results.

**Reviewer Confidence:**

4: Quite sure. I tried to check the important points carefully. It's unlikely, though conceivable, that I missed something that should affect my ratings.

**Typos Grammar Style And Presentation Improvements:**

L437: we provide detailed using -> we provide details using
L1441: replaced by agenerator -> replaced by a generator

Reference at lines 782-784 is missing details.
References at lines 785-792 and 793-798 are the same paper(?).
References at lines 799-805 and 806-810 are the same paper(?).
References at lines 821-828 and 829-835 are the same paper(?).
Reference at lines 875-876 might be better suited as a footnote(?).
References at lines 877-885 and 886-891 are the same paper(?).
References at lines 965-969 and 970-974 are the same paper(?).
References at lines 975-978 and 979-982 are the same paper(?).
Reference at lines 1021-1023 is missing details.
Reference at lines 1061-1063 is missing details.
References at lines 1089-1096 and 1097-1104 are the same paper.
References at lines 1116-1124 and 1125-1133 are the same paper.
Reference at lines 1160-1163 is missing details.
References at lines 1176-1178 and 1208-1210 are the same paper.
References at lines 1269-1274 and 1275-1281 are the same paper.
References at lines 1324-1331 and 1332-1339 are the same paper.

---

> ### Author Rebuttal · Authors · 2023-08-29
>
> We thank RRMe for their review. We are glad to hear they appreciate the living benchmark, the public leaderboard, full access to the code for experiments and that we bring together most open datasets in the legal domain in a single accessible space. We appreciate that they acknowledge our new evidence for a lack of good multilingual legal model and the difficulty of the presented tasks as evidenced by the large potential for improvement.
>
> Reason to reject 1: limited novelty:
> Please refer to our response to reviewer wXJc (Defining a standardized benchmark is incremental, No new methods defined, No novelty in evaluation).
>
> Reason to reject 2: long documents
> As we mentioned in Section 4.3 (lines 461-465), to the best of our knowledge, models capable of handling longer context out of the box are not available multilingually and predominantly trained on English data only. This seems to be the case for SLED also: all the available versions (https://huggingface.co/tau) are pre-trained only on English documents.
> We will expand the introduction (lines 60-79) to further stress this characteristic of legal documents and the current lack of multi-lingual models capable of handling long contexts in literature. We will also include a reference to SLED along the Longformer in Section 4.3.
> We hope that our work will contribute to stimulating the development of multilingual models capable of handling large documents.
>
> Answers to questions:
> A. Limited discussion
> Due to limited space, we kept the discussion short, and we rather focused on the description of the benchmark and the systematic process we followed to create it. We are happy to use the additional page to expand the discussion, including the following points.
>
> B. Why are native monolingual models better than native legal models?
> The results are inconclusive, and we do not see any general trend explaining these phenomena. We studied the papers introducing the native monolingual models and the native legal models. For example, the Romanian legal model had more training data than the native model, but it was out of domain (court decisions) whereas our benchmark data is from legislation in Romania and the EU. On the other hand, the Spanish legal model was trained on approx. 9GB of Spanish legal text and outperformed the native Spanish model (trained on 570GB of Spanish data). We do not have any explanation for this result.
> We will add additional analysis and suggest this to be explored further in future work.
>
> C. How does the jurisdiction influence the results?
> We respectfully disagree with the reviewer. We agree that the jurisdiction dimension is an interesting aspect, but we believe that such analysis is beyond the scope of this work and consciously decided to leave this for future work.
> We believe that studying the effect of the jurisdiction should be done in a single language before it is expanded to multiple languages to properly control for the effect of the language. To achieve this, we would need a dataset that measures the same (or a very similar) task in the same language from different jurisdictions. German text from Switzerland, Austria and Germany, French text from France and Canada or English text from the UK and the US would be potential candidates. Unfortunately, we did not find suitable datasets in the literature. Scaling this experiment up to multilingual datasets and models complicates it significantly.
> We agree that this would be an exciting direction for future research. In particular, it would be interesting to understand whether the jurisdiction information is predominantly learned as part of the LLM or it is instead learned during fine-tuning. Possibly, this could be analyzed by comparing the embedded representation of some keywords inside the LLMs in the different training setting.
> We believe that this research direction would require an entire paper of its own focused on these experiments, and therefore should be considered as a future research direction.
> We will expand our discussion and future works sections to include these considerations.
>
> D. for outcome prediction tasks, documents are written by the court:
> We agree with the reviewer that the judgment prediction task in its classic form should be based on complaints by the parties and not on the facts or even reasoning written by the court. Due to very limited access to the complaint documents, especially multilingually, creating such datasets is extremely challenging. Thus, most recent work used text from court decisions as proxies. However, predicting the judgment outcome based on text written by the court itself can still be a hard task (as evidenced by results on these datasets). Moreover, it may still require legal reasoning capabilities from models because of the need to pick out the correct information. Additionally, we believe that these tasks can also be interesting to conduct post hoc analyses of decisions (e.g., would a model learned over decisions from multiple judges have come to the same conclusion as this specific judge?).
> We want to stress that we based our work on existing literature. Unfortunately, to the best of our knowledge, the number of datasets that also include the material provided by the parties is extremely scarce, especially for languages other than English. Nonetheless, since we designed LEXTREME as a living benchmark, it will be possible to include additional dataset of this type if more will be published in the future.
> As the reviewer noted, we already touched on this limitation. We will use the additional page provided to further expand the discussion and limitation sections with these observations and the comments provided by the reviewer.
>
> E. What is a “better” model?
> Usually, scaling up model size and quality and quality of pretraining data leads to better performance across many benchmarks as demonstrated by mT5 for example (Xue et al., 2021). We see this trend confirmed in LEXTREME (see Figure 1). Surprisingly, monolingual models perform very well on our benchmark (see Table 6).
>
> We are deeply grateful for their detailed list of typos and improvements for the bibliography. We will incorporate these changes for the camera-ready version.

---

### Official Review · Reviewer_wXJc · 2023-08-05

**Soundness:** 3

**Excitement:**

3: Ambivalent: It has merits (e.g., it reports state-of-the-art results, the idea is nice), but there are key weaknesses (e.g., it describes incremental work), and it can significantly benefit from another round of revision. However, I won't object to accepting it if my co-reviewers champion it.

**Paper Topic And Main Contributions:**

This paper introduces LEXTREME, a new multilingual benchmark for evaluating natural language processing (NLP) models on legal and legislative text data. The paper chooses popular models and fine-tune them on the the final benchmark and showcase moderate performance indicating more work needs to be done to improve performance on this benchmark. The key contributions are:

1. The paper compiles 11 existing legal NLP datasets covering 24 languages from 8 language families into a unified benchmark.


2. The work provides baseline results using 5 popular multilingual encoder models (DistilBERT, MiniLM, mDeBERTa, XLM-R base/large) as well as monolingual models.

Overall, this paper makes a valuable contribution to introducing a standardized multilingual benchmark for the legal domain, which has traditionally relied more on monolingual resources. The benchmarks and baselines could aid the future development and evaluation of multilingual legal NLP models.

**Questions For The Authors:**

The paper uses macro F1 for comparison - could micro F1 or label-weighted scores alter the relative model rankings? Did you experiment with other metrics?

**Reasons To Accept:**

Below are the main strengths of the paper:

1. The benchmark would be useful to the community to track multi-lingual progress in the NLP domain.

2. The paper highlights moderate performances of the baseline models and the underperformance of ChatGPT highlighting gaps to be filled in this domain.

**Reasons To Reject:**

The main weakness of the paper is:

1. The work does not provide any model, data, or evaluation novelty and is just a collection of other datasets and models that are already used by everyone in the community. While standardizing the benchmark is a good contribution, the work is just incremental in that all the datasets are already available and people can already evaluate models on these datasets.




**Reproducibility:**

4: Could mostly reproduce the results, but there may be some variation because of sample variance or minor variations in their interpretation of the protocol or method.

**Reviewer Confidence:**

4: Quite sure. I tried to check the important points carefully. It's unlikely, though conceivable, that I missed something that should affect my ratings.

---

> ### Author Rebuttal · Authors · 2023-08-29
>
> We thank wXJc for their review. We are glad to hear that they acknowledge the gap we highlighted by showing moderate performance of baseline models and underperformance of ChatGPT and that they appreciate LEXTREME's usefulness to track multilingual progress in NLP.
>
> Defining a standardized benchmark is incremental: We want to stress that defining a standardized benchmark is not incremental, but a defining contribution in the field, pushing research forward for the following reasons:
> Much like a literature review, equally highly regarded and often highly cited, we also followed a systematic process (described in detail in Section 3.1) analyzing and tagging 108 papers, and finally selecting 11 datasets using clearly defined inclusion and exclusion criteria.
> Standardized benchmarks that also present an aggregation of datasets without providing new methods, such as GLUE (presented at ICLR), SuperGLUE (presented at NeurIPS), and LexGLUE (presented at ACL), are widely used across the field and have been instrumental in shaping NLP progress.
> As evidenced by the acceleration of saturation in standardized NLP benchmarks (https://contextual.ai/plotting-progress-in-ai/), we think that multilinguality and domain-specific subfields of NLP are getting ever more attention since they offer novel challenges for current methods. Our results on LEXTREME show that multiligual legal NLP is still challenging for current models.
>
> No new methods defined: Since we present a standardized benchmark, by itself a large contribution requiring a lengthy description, we did not present new models, since we think this to be out of scope. The goal of our experimental results is only to provide a reference point. Providing this reference point already required significant effort to provide a fair and rigorous evaluation of the current state of the art (689 GPU days spent in total resulting in over 5 months of experiments on our hardware). As indicated by our submission to the Resource and Evaluation track, the focus of our paper lies in the dataset survey, selection, description, and standardization for easy use and fast adoption by the community. LEXTREME is already getting attention from the community for multilingual evaluation in the legal domain, and has also been integrated into HELM.
>
> No novelty in evaluation: Meaningful and fair evaluation and aggregation over multiple dataset configurations and languages is challenging, especially with highly imbalanced label distributions as in LEXTREME. Therefore, we put considerable thought into our evaluation method and present a novel method for fair evaluation in such a scenario. As described in more detail in Section 5, we use macro-averaged F1 score (giving each class equal weight) and aggregate it hierarchically using the harmonic mean, thus promoting models that report good performance robustly across configurations and languages, a property highly desirable in multilinguality in our opinion.
>
> Could other metrics alter model rankings?: Yes, different metrics could alter model rankings, but we specifically focused on the macro averaged score to promote fairness across languages and jurisdictions (as described in detail in Section 5). Including multiple metrics can be misleading and confusing, especially in a benchmark like LEXTREME that includes many configurations and languages. However, we have already computed other metrics, such as micro-F1 and MCC scores to better understand results and we are happy to add them to the appendix for completeness.

---

### Meta-Review · Area_Chair_PoFG · 2023-09-17

**Recommendation:** 4

**Metareview:**

This paper introduces LEXTREME, a new multilingual benchmark for evaluating natural language processing (NLP) models on legal and legislative text data. The paper chooses popular models and fine-tune them on the the final benchmark and showcase moderate performance indicating more work needs to be done to improve performance on this benchmark. The key contributions are:

The paper compiles 11 existing legal NLP datasets covering 24 languages from 8 language families into a unified benchmark.
The work provides baseline results using 5 popular multilingual encoder models (DistilBERT, MiniLM, mDeBERTa, XLM-R base/large) as well as monolingual models.
Overall, this paper makes a valuable contribution to introducing a standardized multilingual benchmark for the legal domain, which has traditionally relied more on monolingual resources. The benchmarks and baselines could aid the future development and evaluation of multilingual legal NLP models.

Reasons To Accept:
- The benchmark would be useful to the community to track multi-lingual progress in the NLP domain.
- The paper highlights moderate performances of the baseline models and the underperformance of ChatGPT highlighting gaps to be filled in this domain.
- This work brings together, in a single accessible space most of the open datasets in the legal domain. This leads to a new multilingual and multi-task benchmark for the legal domain. Furthermore, this is a living benchmark, with a public leaderboard and full access to code to run experiments.


Reasons To Reject:
- The work does not provide any model, data, or evaluation novelty and is just a collection of other datasets and models that are already used by everyone in the community.
- While standardizing the benchmark is a good contribution, the work is just incremental in that all the datasets are already available and people can already evaluate models on these datasets.

---

### Decision · Program_Chairs · 2023-10-07

**Decision:**

Accept-Findings

**Comment:**

This paper introduces LEXTREME, a new multilingual benchmark for evaluating natural language processing (NLP) models on legal and legislative text data. The paper chooses popular models and fine-tune them on the the final benchmark and showcase moderate performance indicating more work needs to be done to improve performance on this benchmark. The key contributions are:

The paper compiles 11 existing legal NLP datasets covering 24 languages from 8 language families into a unified benchmark.
The work provides baseline results using 5 popular multilingual encoder models (DistilBERT, MiniLM, mDeBERTa, XLM-R base/large) as well as monolingual models.
Overall, this paper makes a valuable contribution to introducing a standardized multilingual benchmark for the legal domain, which has traditionally relied more on monolingual resources. The benchmarks and baselines could aid the future development and evaluation of multilingual legal NLP models.

Reasons To Accept:
- The benchmark would be useful to the community to track multi-lingual progress in the NLP domain.
- The paper highlights moderate performances of the baseline models and the underperformance of ChatGPT highlighting gaps to be filled in this domain.
- This work brings together, in a single accessible space most of the open datasets in the legal domain. This leads to a new multilingual and multi-task benchmark for the legal domain. Furthermore, this is a living benchmark, with a public leaderboard and full access to code to run experiments.


Reasons To Reject:
- The work does not provide any model, data, or evaluation novelty and is just a collection of other datasets and models that are already used by everyone in the community.
- While standardizing the benchmark is a good contribution, the work is just incremental in that all the datasets are already available and people can already evaluate models on these datasets.